# VENOMREC: Cross-Modal Interactive Poisoning for Targeted Promotion in Multimodal LLM Recommender Systems

Guowei Guan[1]  Yurong Hao[1]  Jiaming Zhang[1]  Tiantong Wu[1]  Fuyao Zhang[1]  Tianxiang Chen[1]
Longtao Huang[2]  Cyril Leung[1]  Wei Yang Bryan Lim[1]

## Abstract

Multimodal large language models (MLLMs) are pushing recommender systems (RecSys) toward content-grounded retrieval and ranking via cross-modal fusion. We find that while cross-modal consensus often mitigates conventional poisoning that manipulates interaction logs or perturbs a single modality, it also introduces a new attack surface where synchronised multimodal poisoning can reliably steer fused representations along stable semantic directions during fine-tuning. To characterise this threat, we formalise cross-modal interactive poisoning and propose VENOMREC, which performs Exposure Alignment to identify high-exposure regions in the joint embedding space and Cross-modal Interactive Perturbation to craft attention-guided coupled token–patch edits. Experiments on four real-world multimodal datasets demonstrate that VENOMREC consistently outperforms strong baselines, achieving 0.73 mean ER@20 and improving over the strongest baseline by +0.52 absolute ER points on average, while maintaining comparable recommendation utility. Code is available at https://github.com/GuoweiGuan666/VenomRec.

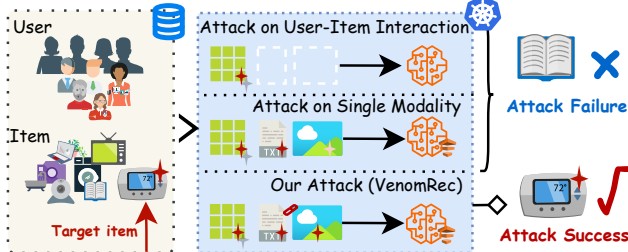

*Figure 1.* Overview of poisoning paradigms on MLLM-RecSys. **Top:** Interaction-level attacks manipulate discrete user-item records (e.g., clicks/ratings) but have limited targeted influence against MLLM-RecSys because they do not directly steer the model's semantic reasoning process. **Middle:** Single-modality attacks perturb either text or image in isolation and are often mitigated by cross-modal consensus in the fusion mechanism, where the unperturbed modality anchors the final semantics. **Bottom:** Our VENOMREC exploits MLLM-RecSys by crafting coupled cross-modal perturbations guided by cross-modal attention.

## 1. Introduction

The rapid evolution of Multimodal Large Language Models (MLLMs) (Wang et al., 2022; Achiam et al., 2023; Geng et al., 2023) is reshaping modern recommender systems (RecSys) (Linden et al., 2003; Gomez-Uribe & Hunt, 2015; Covington et al., 2016) from ID-centric matching engines into content-based architectures capable of cross-modal semantic reasoning. By aligning visual encoders with the la-

tent space of large language models (LLMs) (Alayrac et al., 2022), MLLM-RecSys can jointly interpret an item's textual description and visual appearance, and generate recommendations based on high-dimensional semantic compatibility rather than purely interaction co-occurrence (Geng et al., 2023; Zhang et al., 2025a; Giahi et al., 2025). Crucially, this architectural shift does not necessarily make the system secure; instead, it changes *where* the attack surface lies.

A key empirical observation is that MLLM-RecSys often exhibits *semantic resilience* to existing poisoning paradigms. Concretely, existing attacks designed for conventional RecSys typically manipulate either (i) discrete interaction records (e.g., injecting clicks/ratings) (Wang et al., 2024; Li et al., 2016; Zhang et al., 2020; Hao et al., 2024b;a), or (ii) a single modality (text-only or image-only) (Yang et al., 2023; Liu et al., 2025; Shan et al., 2024). However, when recommendations are driven by cross-modal semantic grounding, such attacks tend to yield limited targeted influence under practical stealth constraints. Interaction-level manipulations (Top in Figure 1) may increase the frequency of a target item in the training phase, yet they do not precisely steer the fused semantic representation that governs content-based retrieval and ranking (Huang et al.,

[1]College of Computing and Data Science, Nanyang Technological University, Singapore [2]Alibaba Group, China. Correspondence to: Yurong Hao <yurong.hao@ntu.edu.sg>.

*Proceedings of the 43rd International Conference on Machine Learning*, Seoul, South Korea. PMLR 306, 2026. Copyright 2026 by the author(s).

2021). Similarly, single-modality perturbations (Middle in Figure 1) are frequently mitigated by cross-modal fusion in practice: when one modality becomes inconsistent, the other modality can serve as a semantic anchor, constraining the final prediction toward a plausible joint interpretation. These observations explain why naive extensions of prior attacks often underperform on MLLM-RecSys.

Paradoxically, the same cross-modal consensus that suppresses independent noise also introduces a new vulnerability. Our investigations suggest that the key security bottleneck lies not in the vulnerability of an individual modality but in the *consensus* mechanism (Dong et al., 2021) established during multimodal fusion. Once an adversary can synchronise manipulations across both modalities, the fusion process may no longer act as a defensive filter; instead, it can amplify the attacker's coordinated signal into a stable semantic direction. This leads to a new threat that is poorly captured by existing attacks.

To expose this vulnerability, we propose VENOMREC (Figure 1, bottom), a framework for cross-modal data poisoning targeting MLLM-RecSys. Specifically, VENOMREC operates by generating coordinated and stealthy perturbations across both visual and textual data. The method first identifies a target "hotspot" direction in the joint embedding space via Exposure Alignment (EA). Then, a novel Cross-modal Interactive Perturbation (CIP) algorithm crafts the poisoned data. CIP leverages the model's cross-modal attention mechanism to identify salient visual patches and textual tokens that are most influential to the fusion process. It then optimises minimal perturbations over these inputs, creating poisoned samples. These samples are designed such that, when the victim model is fine-tuned on the poisoned dataset, the target item's representation is implicitly steered toward the hotspot, thereby increasing its recommendation probability. Extensive experiments demonstrate that VENOMREC consistently yields stronger targeted promotion under strict stealth constraints, surpassing the best competing baseline by an average margin of +0.52 absolute ER@20 and reaching a mean ER@20 of 0.73. Our contributions are summarised as follows:

- We are the first, to our knowledge, to formalise and investigate the threat of cross-modal interactive poisoning attacks specifically targeting Multimodal LLM-based Recommender Systems.

- We propose VENOMREC, a novel attack framework that generates effective and stealthy multimodal perturbations by strategically optimising against the cross-modal attention mechanism.

- We conduct extensive experiments on four real-world multimodal datasets to demonstrate that VENOMREC consistently outperforms single-modality baselines in both attack effectiveness and stealthiness.

## 2. Related Work

**MLLM-RecSys.** Recent advances in MLLMs have reshaped recommender systems from ID-centric matching toward content-grounded architectures by integrating diverse data types, including textual descriptions, visual content, and user behaviour (Geng et al., 2023; Zhang et al., 2025b). A growing body of MLLM-based recommender systems (Wang et al., 2023; Li et al., 2024; Geng et al., 2023; Zhang et al., 2025b; Lin et al., 2024) demonstrates the effectiveness of such multimodal content modelling for recommendation. For example, VIP5 (Geng et al., 2023) proposes a parameter-efficient MLLM framework that unifies vision, language, and personalisation modalities to enhance recommendation tasks. PMMRec (Li et al., 2024) explores a recommender system that relies solely on multi-modal item contents to achieve transferable recommendations. MISS-Rec (Wang et al., 2023) introduces a pre-training framework that learns multimodal, interest-aware sequence representations to enhance recommendation performance. Beyond academic prototypes, NoteLLM-2 (Zhang et al., 2025b) represents a deployed multimodal recommender that integrates vision encoders with large language models to support item-to-item recommendations in real-world platforms. Despite their performance, the security of MLLM-RecSys under cross-modal fusion remains overlooked. In this work, we investigate the threat introduced by this architectural shift.

**Poisoning Attacks against MLLMs.** Data poisoning is a critical security threat where attackers manipulate model behaviour by injecting malicious samples into training data (Biggio et al., 2012; Li et al., 2022; Schwarzschild et al., 2021). It has been explored in image classification (Shafahi et al., 2018), vision-language learning (Carlini & Terzis, 2021; Yang et al., 2023; Liang et al., 2024; 2025), generative models (Shan et al., 2024; Wu et al., 2025), and LLMs (Shu et al., 2023; Ning et al., 2025; 2024). In the recommendation domain in particular, prior work has extensively examined data poisoning against collaborative filtering and sequential recommenders (Wang et al., 2024; Huang et al., 2021; Zhang et al., 2020; Li et al., 2016), primarily through injection of fabricated interaction records. However, these studies primarily focus on single-modality settings. When moving to MLLMs, such conventional paradigms often yield limited influence because the cross-modal grounding in MLLMs acts as a natural defence in practice, where the unperturbed modality serves as a "semantic anchor" to rectify unimodal noise. Shadowcast (Xu et al., 2024) attempted to overcome this by designing "shadow" samples that leverage both visual and textual modalities to enhance both stealthiness and effectiveness. However, it primarily manipulates input-level features to induce latent misalignment, failing to exploit the deep cross-modal fusion mechanisms. This limitation is particularly evident in MLLM-RecSys, where the alignment

of user interactions and cross-modal semantics drives recommendations. Unlike prior works, we turn the cross-modal consensus, originally a defence barrier, into an attack amplifier that biases the model to reach a malicious consensus for effective and stealthy manipulation.

## 3. Problem Formulation

In this section, we formalise the victim MLLM-RecSys framework and define the cross-modal interactive poisoning threat model, encompassing the attacker's knowledge, constraints, and optimisation goals.

### 3.1. Victim MLLM-RecSys Framework

We consider a general MLLM-RecSys framework, exemplified by VIP5 (Geng et al., 2023), which unifies recommendation into a sequence-to-sequence generation paradigm. Let $\mathcal{U}$ and $\mathcal{I}$ denote the sets of users and items. Each item $i \in \mathcal{I}$ is associated with multimodal content consisting of textual description $t_i$ and visual input $v_i$.

**Multimodal Prompt Construction.** For a user $u \in \mathcal{U}$ with interaction history $\mathcal{H}_u = [i_1, \ldots, i_K]$, each item $i_k$ is encoded into a multimodal token sequence: $\mathbf{e}_k = \tau(t_{i_k}) \oplus \zeta(g(v_{i_k}))$, where $\tau(\cdot)$ is the text tokenizer, $g(\cdot)$ is a frozen visual encoder (e.g., CLIP-ViT (Radford et al., 2021)), and $\zeta(\cdot)$ is a trainable projector. The input prompt is constructed:

$$X_u = \mathcal{P} \oplus \tau(u) \oplus [\mathbf{e}_1, \ldots, \mathbf{e}_K], \tag{1}$$

where $\mathcal{P}$ is an instruction template. Crucially, the MLLM's attention layers enable interactions between textual tokens and visual features, serving as the cross-modal fusion mechanism $\Phi_{\boldsymbol{\Theta}}$.

**Training Objective.** Let $\mathcal{D}$ denote the training dataset consisting of multimodal inputs $X$ and their corresponding ground-truth outputs $Y$ (e.g., a sequence of candidate item tokens). We denote the trainable parameters (e.g., adapters and projector $\zeta$) by $\boldsymbol{\Theta}$. The victim model optimises $\boldsymbol{\Theta}$ by minimising a recommendation loss $\mathcal{L}_{\text{rec}}$. A common instantiation is the autoregressive negative log-likelihood (NLL) (Geng et al., 2022):

$$\mathcal{L}_{\text{rec}}(\boldsymbol{\Theta}; \mathcal{D}) = - \mathbb{E}_{(X,Y) \sim \mathcal{D}} \Big[ \sum_{j=1}^{|Y|} \log p_{\boldsymbol{\Theta}}(y_j \mid X, y_{<j}) \Big], \tag{2}$$

where $p_{\boldsymbol{\Theta}}(\cdot)$ is the conditional generation probability predicted by the MLLM, and $y_{<j}$ denotes the tokens generated before step $j$.

### 3.2. Cross-modal Interactive Poisoning

We focus on a targeted poisoning attack aiming to *globally* promote a set of target items $\mathcal{I}^{\dagger} \subseteq \mathcal{I}$ by maximising their exposure across benign users. Unlike conventional attacks on interaction records, we consider a setting where compromised users contribute *user-generated multimodal content* (UGC), such as reviews and images, which are integrated into training via Eq. (1). To formally describe this coupled attack behaviour and its constraints, we define cross-modal interactive poisoning as follows.

**Definition 3.1** (Cross-modal Interactive Poisoning). Given a victim $f_{\boldsymbol{\Theta}}$ with a fusion operator $\Phi_{\boldsymbol{\Theta}}$, the attack is characterised as a tuple $\mathcal{A} = (\mathcal{U}_{\text{mal}}, \mathcal{I}^{\dagger}, \mathcal{T})$, where:

- $\mathcal{U}_{\text{mal}} \subseteq \mathcal{U}$ is the set of compromised users with $|\mathcal{U}_{\text{mal}}| \leq \rho|\mathcal{U}|$, where $\rho$ is the compromised-user ratio;

- $\mathcal{I}^{\dagger}$ is the set of target items for global promotion;

- $\mathcal{T} : (t, v) \mapsto (\tilde{t}, \tilde{v})$ is a coupled transformation mapping clean UGC to poisoned pairs within a stealthiness manifold $\mathcal{B}$.

The transformation $\mathcal{T}$ is *coupled* such that perturbations in $t$ and $v$ are jointly determined with respect to the fusion operator $\Phi_{\boldsymbol{\Theta}}$ to maximise target exposure in the post-poisoning state $f_{\boldsymbol{\Theta}^*}$, where $\boldsymbol{\Theta}^*$ is optimised on the combined dataset $\tilde{\mathcal{D}} = \mathcal{D}_{\text{ben}} \cup \tilde{\mathcal{D}}_{\text{mal}}$.

**Stealthiness Constraints $\mathcal{B}$.** $\mathcal{B}$ constrains poisoned samples to satisfy: (i) *unimodal naturalness*, ensuring each modality remains independently plausible; and (ii) *cross-modal coherence*, ensuring the modified text and image remain semantically consistent after fusion via $\Phi_{\boldsymbol{\Theta}}$.

**Attacker's Knowledge and Capabilities.** We assume a realistic grey-box setting where the adversary lacks access to the victim's private parameters $\boldsymbol{\Theta}$ or gradients. However, the adversary is aware of the publicly available backbones (e.g., CLIP, T5) which serve as proxies for estimating $\Phi_{\boldsymbol{\Theta}}$ and token–patch correspondences. Within $\mathcal{U}_{\text{mal}}$, the adversary can manipulate interaction histories and UGC subject to the budget $|\mathcal{U}_{\text{mal}}| \leq \rho|\mathcal{U}|$ and stealthiness $\mathcal{B}$.

**Attacker's Goal.** The adversary aims to craft the poisoned subset $\tilde{\mathcal{D}}_{\text{mal}}$ to maximise the exposure of the target item(s). Let $\mathcal{R}(\mathcal{I}^{\dagger}; \boldsymbol{\Theta})$ denote a target exposure objective, e.g.,

$$\mathcal{R}(\mathcal{I}^{\dagger}; \boldsymbol{\Theta}) = \frac{1}{|\mathcal{I}^{\dagger}|} \sum_{i^{\dagger} \in \mathcal{I}^{\dagger}} \mathbb{E}_{u \sim \mathcal{U} \setminus \mathcal{U}_{\text{mal}}} \big[ \mathbb{I}(i^{\dagger} \in \text{Top-}K(u; \boldsymbol{\Theta})) \big].$$

Formally, this cross-modal interactive poisoning attack is modelled as the following optimisation:

$$\max_{\tilde{\mathcal{D}}_{\text{mal}}} \quad \mathcal{R}(\mathcal{I}^{\dagger}; \boldsymbol{\Theta}^*)$$
$$\text{s.t.} \quad (\tilde{t}_{u,i^{\dagger}}, \tilde{v}_{u,i^{\dagger}}) \in \mathcal{B}, \ \forall u \in \mathcal{U}_{\text{mal}}, \ \forall i^{\dagger} \in \mathcal{I}^{\dagger}, \tag{3}$$
$$\boldsymbol{\Theta}^* = \arg\min_{\boldsymbol{\Theta}} \ \mathcal{L}_{\text{rec}}(\boldsymbol{\Theta}; \mathcal{D}_{\text{ben}} \cup \tilde{\mathcal{D}}_{\text{mal}}).$$

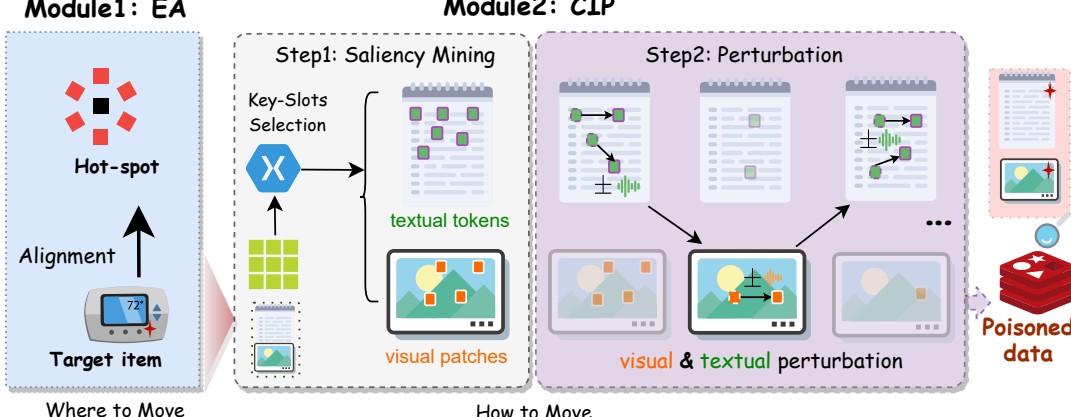

*Figure 2.* The schematic overview of VENOMREC. The framework operates in two strategic phases: (1) **Exposure Alignment (EA)** constructs a latent target centroid from high-exposure anchors to guide the attack direction. (2) **Cross-modal Interactive Perturbation (CIP)** leverages attention-guided saliency to optimise visual features and textual tokens iteratively.

The inner problem reflects the victim's standard training on poisoned data, while the outer problem captures the adversary's strategic promotion objective by crafting $\hat{\mathcal{D}}_{\mathrm{mal}}$. Note that while poisoning is applied only to compromised users $u \in \mathcal{U}_{\mathrm{mal}}$, the exposure objective is evaluated over benign users $u \in \mathcal{U} \setminus \mathcal{U}_{\mathrm{mal}}$ to reflect global promotion.

## 4. Our Attack: VENOMREC

In this section, we present VENOMREC, a novel cross-modal interactive poisoning attack. As established in Def. 3.1, VENOMREC operationalises the coupled transformation $\mathcal{T}$ through a two-stage strategy: (1) Exposure Alignment (EA), which determines *where to move* the target representation, and (2) Cross-modal Interactive Perturbation (CIP), which determines *how to move* it via iterative co-adaptation.

### 4.1. Method Overview

The adversary's objective is to craft poisoned UGC pairs $(\tilde{t}_{u,i^{\dagger}}, \tilde{v}_{u,i^{\dagger}})$ that steer the fused representation toward a high-exposure semantic region while satisfying the stealthiness manifold $\mathcal{B}$. Directly solving the bilevel optimisation in Eq. 3 is intractable due to the unknown $\mathcal{D}_{\mathrm{ben}}$ and the private parameters $\boldsymbol{\Theta}$. To overcome this, VENOMREC utilises a surrogate-based framework. We employ publicly available pretrained backbones (e.g., CLIP and T5) to instantiate a multimodal encoder $\phi(\cdot)$, which serves as a usable proxy for the victim's fusion operator $\Phi_{\boldsymbol{\Theta}}$. Figure 2 illustrates the workflow of VENOMREC. Specifically, it consists of two modules: (1) EA identifies a high-exposure hotspot in the joint embedding space and constructs a latent centroid, which serves as an attack destination; and (2) CIP leverages cross-modal attention to locate the most influential token–patch correspondences that dominate fusion and iteratively optimises coordinated edits on text and image to steer the fused representation toward the hotspot while preserving

unimodal naturalness and cross-modal coherence. Finally, the adversary injects the poisoned target item into the interaction histories of compromised users to form $\hat{\mathcal{D}}_{\mathrm{mal}}$. See Algorithm 1 in the Appendix for the full procedure.

### 4.2. Module I: Exposure Alignment

The goal of EA is to instantiate the abstract high-exposure region into a concrete target representation $\mathbf{z}^{\star}$ that guides the attack destination.

**High-Exposure Anchor Mining.** We hypothesise that naturally high-exposure items share latent semantic characteristics that are relatively stable across model instantiations. This is motivated by the fact that publicly observable exposure/popularity signals often reflect persistent user preferences and platform-level traffic allocation, making their semantic patterns less sensitive to specific model parameters. Accordingly, we mine a set of high-exposure anchor items $\mathcal{H}' \subseteq \mathcal{I}$ from publicly observable popularity lists (e.g., "Best Sellers") and retain only anchors within the same category as the target item $i^{\dagger}$ to ensure semantic compatibility.

**Centroid-based Destination.** Let $\mathbf{z}_j = \phi(t_j, v_j)$ denote the proxy fused representation of an anchor item $j \in \mathcal{H}'$. We define the high-exposure centroid as:

$$\mathbf{z}^{\star} \triangleq \mathrm{Norm}\left(\frac{1}{|\mathcal{H}'|} \sum_{j \in \mathcal{H}'} \mathbf{z}_j\right), \quad (4)$$

where $\mathrm{Norm}(\cdot)$ denotes $\ell_2$ normalisation. This centroid $\mathbf{z}^{\star}$ serves as the attack destination for steering the target item. To guide subsequent perturbation, we define a latent alignment objective that encourages the target embedding to move toward the hotspot:

$$\mathcal{L}_{\mathrm{adv}}(\tilde{t}_{u,i^{\dagger}}, \tilde{v}_{u,i^{\dagger}}) = 1 - \cos\left(\phi(\tilde{t}_{u,i^{\dagger}}, \tilde{v}_{u,i^{\dagger}}), \mathbf{z}^{\star}\right). \quad (5)$$

Notably, EA does not require any access to the victim training loop; it only provides a destination that captures the high-exposure area in the joint embedding space.

### 4.3. Module II: Cross-modal Interactive Perturbation

Given attack direction $\mathbf{z}^\star$, CIP determines *how to move* the target item by iteratively co-adapting both modalities to exploit the interdependency inherent in cross-modal fusion.

**Attention-Guided Saliency Mining.** Instead of injecting independent noise, CIP leverages the proxy model's cross-modal attention to identify fusion-sensitive token–patch correspondences. Let $\mathbf{A} \in \mathbb{R}^{L_t \times L_v}$ denote the aggregated cross-modal attention matrix between $L_t$ textual tokens and $L_v$ visual patches, obtained by combining multi-head and multi-layer attentions via attention rollout. We define sensitivity scores for token position $\ell$ and patch index $p$ as:

$$s_\ell = \frac{1}{L_v} \sum_{j=1}^{L_v} \mathbf{A}_{\ell,j}, \quad s_p = \frac{1}{L_t} \sum_{i=1}^{L_t} \mathbf{A}_{i,p}. \tag{6}$$

We construct binary perturbation masks $\mathbf{m}_{\mathrm{txt}}$ and $\mathbf{m}_{\mathrm{vis}}$ by selecting the top-$k$ elements with the highest sensitivity. Crucially, these masks are *recomputed* after each update round to capture the shift in fusion sensitivity, enforcing the *coupled* nature of the transformation $\mathcal{T}$.

**Visual Perturbation.** For the visual modality, we update the selected patch-level features to steer the fused representation toward $\mathbf{z}^\star$ under a strict budget. Denote the current visual representation by $\mathbf{v}$ (e.g., projected patch features), and the original representation by $\mathbf{v}_{\mathrm{orig}}$. For the masked dimensions, we apply a directional sign projection. The update rule at iteration $r$ is:

$$\mathbf{v}_{r+1} = \Pi_\epsilon^\infty \left( \mathbf{v}_r + \eta \cdot \mathrm{sign} \left( \mathbf{d}_r \odot \mathbf{m}_{\mathrm{vis}} \right) \right), \tag{7}$$

where $\eta$ is the step size, $\odot$ denotes element-wise masking, and $\Pi_\epsilon^\infty$ is a projection operator that clips the values to the range $[\mathbf{v}_{\mathrm{orig}} - \epsilon, \mathbf{v}_{\mathrm{orig}} + \epsilon]$. Here $\mathbf{d}_r$ denotes a direction vector estimated without gradients by probing a small set of candidate perturbations on the masked patches and selecting the one that yields the largest increase in $\cos(\phi(\tilde{t}_{u,i^\dagger}, \tilde{v}_{u,i^\dagger}), \mathbf{z}^\star)$. This effectively enforces a dimension-wise $\ell_\infty$ constraint, ensuring the perturbation remains visually inconspicuous while statistically aligning with the high-exposure manifold.

**Textual Perturbation.** For the textual modality, we optimise discrete tokens under linguistic and semantic constraints. For token positions selected by $\mathbf{m}_{\mathrm{txt}}$, we sample a candidate set of replacements and apply greedy search to select the edit that best improves the alignment to $\mathbf{z}^\star$ while preserving text fluency. To ensure cross-modal coherence, we further reject candidates that significantly reduce the proxy image–text semantic consistency between the edited text and the current visual input. These constraints collectively implement the stealthiness set $\mathcal{B}$, enforcing both unimodal naturalness and cross-modal coherence.

**Interactive Co-adaptation Loop.** CIP alternates between the visual update (Eq. 7) and the textual perturbation for $R$ rounds. After each iteration, the proxy attention map $\mathbf{A}$ is recomputed to capture the shifted fusion sensitivity, and the masks $\mathbf{m}_{\mathrm{txt}}, \mathbf{m}_{\mathrm{vis}}$ are updated accordingly to maintain the interactive nature of the attack. The loop terminates when the alignment score $\cos(\phi(\tilde{t}_{u,i^\dagger}, \tilde{v}_{u,i^\dagger}), \mathbf{z}^\star)$ exceeds a predefined threshold or the iteration budget $R$ is exhausted. The resulting poisoned UGC pairs are then used to construct the final malicious subset $\tilde{\mathcal{D}}_{\mathrm{mal}}$.

## 5. Experimental Evaluation

### 5.1. Experiment Settings

**Datasets and Victim Models.** We conduct experiments on three widely-used Amazon product review datasets (Clothing, Sports, and Toys (McAuley & Leskovec, 2013)) and a large-scale short-video dataset MicroLens (Ni et al., 2025), with detailed statistics summarised in Table 1.

*Table 1.* Dataset information.

| Dataset | Clothing | Sports | Toys | MicroLens |
|---|---|---|---|---|
| #User | 39,387 | 35,598 | 19,412 | 100,000 |
| #Item | 23,033 | 18,357 | 11,924 | 19,738 |
| #Image | 22,299 | 17,943 | 11,895 | 19,738 |
| #Review | 278,677 | 296,337 | 167,597 | 719,405 |
| #Interaction | 278,677 | 296,337 | 167,597 | 719,405 |
| Density (%) | 0.0307 | 0.0453 | 0.0724 | 0.0364 |

These datasets cover both e-commerce and short-video domains, providing rich multimodal content (images, text, and user interaction histories) with varying degrees of sparsity, enabling a rigorous evaluation of MLLM-based recommenders. Following established protocols (Zhang et al., 2024), we utilise pre-extracted CLIP (ViT-B/32) visual embeddings (Radford et al., 2021) and adopt the official 8:1:1 split for training, validation, and testing. Our primary victim model is **VIP5** (Geng et al., 2023), a state-of-the-art multimodal recommender built on the frozen T5-small backbone (Raffel et al., 2020); we also include a scaled-up T5-base variant for scaling analysis. To further assess attack transferability across model architectures, we additionally evaluate VENOMREC on **DiffMM** (Jiang et al., 2024), a traditional multimodal recommender. VIP5 is fine-tuned using Parameter-Efficient Fine-Tuning (PEFT) via lightweight adapters (Sung et al., 2022), while DiffMM is trained following its original protocol.

**Baselines.** We evaluate VENOMREC against ten representative baselines across three categories:
(1) **Interaction-level**: DirectBoost (Lam & Riedl, 2004),

RandomAttack (O'Mahony et al., 2005), and PopularAttack (Zhou et al., 2015), which manipulate user-item interactions without content modification.

(2) **Unimodal**: We include both textual- and visual-based attack baselines. Textual attacks include TextFooler (Jin et al., 2020), DeepwordBug (Gao et al., 2018), PuncAttack (Formento et al., 2023), and BERTAttack (Li et al., 2020); and visual attacks include INSA (Liu & Larson, 2021) and EXPA (Liu & Larson, 2021).

(3) **Multimodal**: Shadowcast (Xu et al., 2024), which generates adversarial examples targeting MLLMs.

**Evaluation Metrics.** We conduct a comprehensive evaluation of VENOMREC in terms of attack effectiveness, stealthiness, and recommendation utility.

(1) **Attack effectiveness** is measured by Exposure Rate (ER@$K$), the proportion of test users for whom the target item appears in the top-$K$ recommendations.

(2) **Attack stealthiness** is assessed via two sub-dimensions. Unimodal Naturalness, measured by ROUGE scores (textual fluency) (Lin, 2004) and FID (visual fidelity) (Heusel et al., 2017); and Cross-modal Coherence, measured by Semantic Alignment and Latent Direction Alignment, to ensure synchronised perturbations.

(3) **Recommendation utility** is evaluated by standard ranking metrics, Hit Ratio (HR@$K$) and Normalised Discounted Cumulative Gain (NDCG@$K$), computed on benign users to measure model performance.

**Implementation Details.** We implement all models using the PyTorch framework and optimise them via the AdamW optimiser (Loshchilov & Hutter, 2017). To ensure a rigorous and fair comparison, VENOMREC and all baselines are fine-tuned starting from the identical pre-trained checkpoint. We maintain consistent hyperparameters and training configurations across all experiments, including a learning rate of $1 \times 10^{-3}$, a batch size of 128, a compromised user ratio of 0.1%, and standardised instruction templates, so as to isolate the impact of the poisoning strategy. Unless stated otherwise, target items are randomly sampled from the bottom 20% of items ranked by interaction frequency, simulating a challenging cold-start promotion scenario. All experiments are conducted on NVIDIA L20 GPUs.

Due to space constraints, additional details on baselines and metrics are in Appendix A.2. We primarily evaluate VIP5 using the T5-small backbone, with extended results for the T5-base version provided in Appendix Table 6. Results on the MicroLens dataset are reported in Appendix A.7. Transferability evaluations against **DiffMM** (Jiang et al., 2024) are also deferred to Appendix Table 8.

## 5.2. Overall Performance Evaluation

**Attack Effectiveness.** We evaluate the targeted promotion performance of VENOMREC against all baselines across three Amazon datasets under both Few-Shot and Zero-Shot settings (Brown et al., 2020). The results, summarised in Table 2, reveal three key findings. **First**, VENOMREC achieves a substantial performance advantage over existing methods in attack effectiveness. On the Clothing dataset (Few-Shot), while traditional Interaction-level attacks (e.g., DirectBoost, PopularAttack) and semantic perturbations (e.g., TextFooler) yield negligible exposure gains (ER@5 <0.01), VENOMREC achieves an ER@5 of 0.1158. The performance gap becomes even more pronounced at broader cutoffs: at Top-20, VENOMREC attains a dominant exposure rate of 0.7170. Crucially, these gains are consistent across the Sports and Toys datasets, indicating that VENOMREC reliably steers representations into high-exposure latent regions regardless of the specific product domain. **Second**, VENOMREC demonstrates robust generalisation in Zero-Shot scenarios. Remarkably, on the Clothing dataset, the attack effectiveness of VENOMREC improves in the Zero-Shot setting (ER@20=0.7317) compared to the Few-Shot setting. This suggests that our EA module captures universal, high-exposure semantic prototypes that remain effective even without specific fine-tuning data, whereas baseline methods often struggle to generalise in the absence of dense interaction signals. **Third**, cross-modal interaction is critical for circumventing fusion-based defences. Single-modality attacks, such as the vision-only EXPA (ER@20=0.1252) and text-only TextFooler (ER@20=0.0211), fail to achieve significant promotion. This validates our hypothesis regarding the inherent semantic resilience of MLLM-RecSys; perturbing a single modality allows the fusion mechanism to rectify the inconsistency using the unperturbed modality. In contrast, VENOMREC's synchronised perturbation strategy effectively bypasses this consensus check, leading to reliable item promotion.

**Attack Stealthiness and Recommendation Utility.** We further examine whether the substantial gains in target exposure are achieved at the cost of recommendation quality degradation or perceptible artefacts. First, **in terms of recommendation utility**, the results in Table 2 indicate that VENOMREC operates in a targeted and controlled manner. Both HR@K and NDCG@K remain comparable to the clean (*NoAttack*) baseline across all datasets, suggesting that the proposed attack elevates the exposure of specific targets without adversely affecting the overall ranking quality experienced by benign users. Second, **with respect to attack stealthiness**, Table 3 shows that VENOMREC satisfies the dual requirements of unimodal naturalness and cross-modal coherence. For unimodal naturalness, the generated textual perturbations achieve ROUGE-L scores consistently around or above 90.0, while the visual perturbations maintain low FID scores, indicating that both modalities remain linguistically fluent and visually plausible. More importantly, the cross-modal coherence metrics provide direct evidence supporting our core hypothesis. Compared to

*Table 2.* Main results of attack effectiveness and recommendation utility. We compare VENOMREC against various baselines on Clothing, Sports, and Toys datasets under Few-Shot and Zero-Shot settings. The best performance in each category is bolded and underlined.

**Few-Shot (Clothing)**

| Attack | Top-5 HR | Top-5 NDCG | Top-5 ER | Top-10 HR | Top-10 NDCG | Top-10 ER | Top-20 HR | Top-20 NDCG | Top-20 ER |
|---|---|---|---|---|---|---|---|---|---|
| NoAttack | 0.1439 | 0.0989 | 0 | 0.2225 | 0.1242 | 0 | 0.3421 | 0.1543 | 0 |
| DirectBoost | 0.1376 | 0.0935 | 0.0034 | 0.2175 | 0.1191 | 0.0047 | 0.3341 | 0.1484 | 0.0223 |
| RandomAttack | 0.1364 | 0.0925 | 0.0081 | 0.2170 | 0.1183 | 0.0155 | 0.3357 | 0.1482 | 0.1331 |
| PopularAttack | 0.1386 | 0.0943 | 0.0033 | 0.2178 | 0.1196 | 0.0049 | 0.3330 | 0.1486 | 0.0429 |
| TextFooler | 0.1433 | 0.0988 | 0.0032 | 0.2214 | 0.1237 | 0.0041 | 0.3362 | 0.1527 | 0.0211 |
| DeepwordBug | 0.1386 | 0.0948 | 0.0034 | 0.2156 | 0.1195 | 0.0064 | 0.3343 | 0.1493 | 0.0567 |
| PuncAttack | 0.1416 | 0.0965 | 0.0018 | 0.2199 | 0.1215 | 0.0033 | 0.3347 | 0.1504 | 0.0230 |
| BERTAttack | 0.1375 | 0.0942 | 0.0003 | 0.2141 | 0.1187 | 0.0009 | 0.3310 | 0.1481 | 0.0112 |
| INSA | 0.1431 | 0.0975 | 0.0048 | 0.2201 | 0.1221 | 0.0064 | 0.3379 | 0.1518 | 0.0871 |
| EXPA | 0.1419 | 0.0974 | 0.0060 | 0.2192 | 0.1222 | 0.0094 | 0.3345 | 0.1512 | 0.1252 |
| Shadowcast | 0.1404 | 0.0962 | 0.0041 | 0.2156 | 0.1203 | 0.0073 | 0.3319 | 0.1496 | 0.0472 |
| **VENOMREC** | 0.1402 | 0.0948 | **0.1158** | 0.2199 | 0.1204 | **0.2596** | 0.3377 | 0.1500 | **0.7170** |

**Zero-Shot (Clothing)**

| Attack | Top-5 HR | Top-5 NDCG | Top-5 ER | Top-10 HR | Top-10 NDCG | Top-10 ER | Top-20 HR | Top-20 NDCG | Top-20 ER |
|---|---|---|---|---|---|---|---|---|---|
| NoAttack | 0.1437 | 0.0985 | 0 | 0.2218 | 0.1236 | 0 | 0.3386 | 0.1530 | 0 |
| DirectBoost | 0.1372 | 0.0931 | 0.0033 | 0.2156 | 0.1183 | 0.0045 | 0.3343 | 0.1482 | 0.0212 |
| RandomAttack | 0.1344 | 0.0912 | 0.0068 | 0.2160 | 0.1173 | 0.0122 | 0.3357 | 0.1474 | 0.1198 |
| PopularAttack | 0.1381 | 0.0941 | 0.0035 | 0.2163 | 0.1192 | 0.0054 | 0.3333 | 0.1487 | 0.0454 |
| TextFooler | 0.1436 | 0.0983 | 0.0036 | 0.2227 | 0.1236 | 0.0044 | 0.3367 | 0.1523 | 0.0217 |
| DeepwordBug | 0.1376 | 0.0938 | 0.0029 | 0.2160 | 0.1189 | 0.0055 | 0.3337 | 0.1485 | 0.0511 |
| PuncAttack | 0.1402 | 0.0956 | 0.0023 | 0.2183 | 0.1205 | 0.0040 | 0.3346 | 0.1498 | 0.0214 |
| BERTAttack | 0.1371 | 0.0929 | 0.0001 | 0.2111 | 0.1166 | 0.0005 | 0.3302 | 0.1466 | 0.0068 |
| INSA | 0.1405 | 0.0958 | 0.0043 | 0.2205 | 0.1214 | 0.0062 | 0.3365 | 0.1506 | 0.0779 |
| EXPA | 0.1396 | 0.0955 | 0.0055 | 0.2184 | 0.1207 | 0.0088 | 0.3342 | 0.1499 | 0.1247 |
| Shadowcast | 0.1405 | 0.0957 | 0.0032 | 0.2152 | 0.1196 | 0.0066 | 0.3322 | 0.1491 | 0.0451 |
| **VENOMREC** | 0.1402 | 0.0945 | **0.1337** | 0.2206 | 0.1203 | **0.2820** | 0.3381 | 0.1499 | **0.7317** |

**Few-Shot (Sports)**

| Attack | Top-5 HR | Top-5 NDCG | Top-5 ER | Top-10 HR | Top-10 NDCG | Top-10 ER | Top-20 HR | Top-20 NDCG | Top-20 ER |
|---|---|---|---|---|---|---|---|---|---|
| NoAttack | 0.1797 | 0.1237 | 0 | 0.2600 | 0.1495 | 0 | 0.3771 | 0.1790 | 0 |
| DirectBoost | 0.1690 | 0.1170 | 0.0001 | 0.2447 | 0.1413 | 0.0011 | 0.3549 | 0.1690 | 0.0752 |
| RandomAttack | 0.1670 | 0.1158 | 0.0003 | 0.2436 | 0.1404 | 0.0016 | 0.3520 | 0.1676 | 0.1194 |
| PopularAttack | 0.1680 | 0.1160 | 0.0001 | 0.2437 | 0.1402 | 0.0008 | 0.3559 | 0.1685 | 0.0679 |
| TextFooler | 0.1693 | 0.1170 | 0.0001 | 0.2434 | 0.1408 | 0.0004 | 0.3564 | 0.1692 | 0.0677 |
| DeepwordBug | 0.1678 | 0.1165 | 0.0003 | 0.2430 | 0.1406 | 0.0014 | 0.3547 | 0.1687 | 0.0777 |
| PuncAttack | 0.1647 | 0.1145 | 0.0002 | 0.2394 | 0.1385 | 0.0006 | 0.3520 | 0.1668 | 0.0624 |
| BERTAttack | 0.1673 | 0.1163 | 0 | 0.2418 | 0.1402 | 0.0001 | 0.3561 | 0.1689 | 0.0541 |
| INSA | 0.1696 | 0.1171 | 0.0008 | 0.2444 | 0.1411 | 0.0046 | 0.3544 | 0.1688 | 0.1653 |
| EXPA | 0.1696 | 0.1167 | 0.0002 | 0.2455 | 0.1409 | 0.0007 | 0.3547 | 0.1684 | 0.0880 |
| Shadowcast | 0.1828 | 0.1256 | 0.0004 | 0.2657 | 0.1523 | 0.0029 | 0.3767 | 0.1803 | 0.0645 |
| **VENOMREC** | 0.1699 | 0.1178 | **0.0846** | 0.2457 | 0.1421 | **0.1746** | 0.3569 | 0.1701 | **0.5837** |

**Zero-Shot (Sports)**

| Attack | Top-5 HR | Top-5 NDCG | Top-5 ER | Top-10 HR | Top-10 NDCG | Top-10 ER | Top-20 HR | Top-20 NDCG | Top-20 ER |
|---|---|---|---|---|---|---|---|---|---|
| NoAttack | 0.1778 | 0.1224 | 0 | 0.2585 | 0.1483 | 0 | 0.3766 | 0.1781 | 0 |
| DirectBoost | 0.1720 | 0.1177 | 0.0003 | 0.2481 | 0.1422 | 0.0018 | 0.3610 | 0.1706 | 0.1120 |
| RandomAttack | 0.1682 | 0.1166 | 0.0004 | 0.2464 | 0.1416 | 0.0022 | 0.3592 | 0.1700 | 0.1245 |
| PopularAttack | 0.1714 | 0.1181 | 0.0004 | 0.2485 | 0.1429 | 0.0015 | 0.3626 | 0.1716 | 0.0837 |
| TextFooler | 0.1705 | 0.1179 | 0 | 0.2465 | 0.1423 | 0.0004 | 0.3632 | 0.1716 | 0.0660 |
| DeepwordBug | 0.1720 | 0.1186 | 0.0004 | 0.2466 | 0.1425 | 0.0015 | 0.3640 | 0.1720 | 0.0818 |
| PuncAttack | 0.1703 | 0.1174 | 0.0002 | 0.2467 | 0.1419 | 0.0008 | 0.3615 | 0.1708 | 0.0604 |
| BERTAttack | 0.1706 | 0.1179 | 0 | 0.2455 | 0.1419 | 0.0002 | 0.3634 | 0.1716 | 0.0503 |
| INSA | 0.1732 | 0.1188 | 0.0010 | 0.2502 | 0.1435 | 0.0050 | 0.3613 | 0.1715 | 0.1710 |
| EXPA | 0.1716 | 0.1183 | 0.0001 | 0.2495 | 0.1434 | 0.0009 | 0.3625 | 0.1718 | 0.0859 |
| Shadowcast | 0.1820 | 0.1250 | 0.0001 | 0.2657 | 0.1519 | 0.0001 | 0.3768 | 0.1799 | 0.0575 |
| **VENOMREC** | 0.1717 | 0.1190 | **0.0838** | 0.2479 | 0.1435 | **0.1731** | 0.3636 | 0.1726 | **0.5825** |

**Few-Shot (Toys)**

| Attack | Top-5 HR | Top-5 NDCG | Top-5 ER | Top-10 HR | Top-10 NDCG | Top-10 ER | Top-20 HR | Top-20 NDCG | Top-20 ER |
|---|---|---|---|---|---|---|---|---|---|
| NoAttack | 0.1452 | 0.0963 | 0 | 0.2238 | 0.1215 | 0 | 0.3462 | 0.1523 | 0 |
| DirectBoost | 0.1363 | 0.0922 | 0.0004 | 0.2082 | 0.1152 | 0.0029 | 0.3234 | 0.1441 | 0.0917 |
| RandomAttack | 0.1297 | 0.0882 | 0.0004 | 0.2005 | 0.1109 | 0.0025 | 0.3200 | 0.1408 | 0.0618 |
| PopularAttack | 0.1293 | 0.0885 | 0.0001 | 0.2026 | 0.1120 | 0.0013 | 0.3193 | 0.1412 | 0.0442 |
| TextFooler | 0.1447 | 0.0973 | 0.0100 | 0.2234 | 0.1225 | 0.0219 | 0.3450 | 0.1531 | 0.1217 |
| DeepwordBug | 0.1459 | 0.0970 | 0.0047 | 0.2244 | 0.1221 | 0.0122 | 0.3486 | 0.1533 | 0.0712 |
| PuncAttack | 0.1414 | 0.0953 | 0.0077 | 0.2196 | 0.1204 | 0.0177 | 0.3472 | 0.1524 | 0.0866 |
| BERTAttack | 0.1461 | 0.0975 | 0.0076 | 0.2251 | 0.1228 | 0.0168 | 0.3473 | 0.1535 | 0.0808 |
| INSA | 0.1472 | 0.0986 | 0.0175 | 0.2278 | 0.1244 | 0.0432 | 0.3505 | 0.1552 | 0.2579 |
| EXPA | 0.1458 | 0.0982 | 0.0177 | 0.2248 | 0.1235 | 0.0514 | 0.3470 | 0.1543 | 0.3200 |
| Shadowcast | 0.1452 | 0.0970 | 0.0027 | 0.2254 | 0.1226 | 0.0078 | 0.3507 | 0.1541 | 0.0570 |
| **VENOMREC** | 0.1341 | 0.0910 | **0.4065** | 0.2059 | 0.1139 | **0.6229** | 0.3197 | 0.1425 | **0.8674** |

**Zero-Shot (Toys)**

| Attack | Top-5 HR | Top-5 NDCG | Top-5 ER | Top-10 HR | Top-10 NDCG | Top-10 ER | Top-20 HR | Top-20 NDCG | Top-20 ER |
|---|---|---|---|---|---|---|---|---|---|
| NoAttack | 0.1433 | 0.0957 | 0 | 0.2211 | 0.1207 | 0 | 0.3438 | 0.1515 | 0 |
| DirectBoost | 0.1420 | 0.0953 | 0.0020 | 0.2192 | 0.1201 | 0.0103 | 0.3392 | 0.1501 | 0.1117 |
| RandomAttack | 0.1381 | 0.0931 | 0.0012 | 0.2110 | 0.1165 | 0.0063 | 0.3337 | 0.1473 | 0.0787 |
| PopularAttack | 0.1374 | 0.0935 | 0.0014 | 0.2146 | 0.1183 | 0.0051 | 0.3344 | 0.1484 | 0.0639 |
| TextFooler | 0.1426 | 0.0956 | 0.0120 | 0.2229 | 0.1213 | 0.0334 | 0.3432 | 0.1516 | 0.1995 |
| DeepwordBug | 0.1427 | 0.0959 | 0.0036 | 0.2223 | 0.1214 | 0.0120 | 0.3472 | 0.1528 | 0.0749 |
| PuncAttack | 0.1409 | 0.0954 | 0.0065 | 0.2189 | 0.1204 | 0.0158 | 0.3430 | 0.1515 | 0.0797 |
| BERTAttack | 0.1466 | 0.0979 | 0.0069 | 0.2230 | 0.1224 | 0.0160 | 0.3431 | 0.1526 | 0.0799 |
| INSA | 0.1453 | 0.0971 | 0.0168 | 0.2233 | 0.1222 | 0.0414 | 0.3479 | 0.1535 | 0.2405 |
| EXPA | 0.1454 | 0.0974 | 0.0178 | 0.2226 | 0.1222 | 0.0558 | 0.3446 | 0.1529 | 0.3344 |
| Shadowcast | 0.1434 | 0.0960 | 0.0021 | 0.2246 | 0.1219 | 0.0067 | 0.3491 | 0.1532 | 0.0537 |
| **VENOMREC** | 0.1397 | 0.0943 | **0.4316** | 0.2188 | 0.1195 | **0.6442** | 0.3367 | 0.1491 | **0.8781** |

the baseline, VENOMREC exhibits substantially smaller Semantic Consistency deviations (e.g., 0.0033 vs. 0.0306 on Clothing) together with positive Latent Direction Alignment. These results indicate that, rather than introducing unstructured or modality-isolated noise, VENOMREC constructs coordinated perturbations in which visual and textual representations are adjusted along consistent semantic directions. Such coordination preserves intrinsic image–text correlations, enabling the poisoned samples to bypass fusion-based consensus mechanisms and effectively influence the model's fused representations while remaining difficult to detect.

### 5.3. Impact of Compromised User Ratio

We investigate the scalability and cost–benefit trade-offs of VENOMREC by varying the compromised user ratios, $\rho \in$

*Table 3.* Stealthiness across different datasets.

| Attack | Unimodal Naturalness | | | | Cross-modal Coherence | |
|---|---|---|---|---|---|---|
| | ROUGE-1 | ROUGE-2 | ROUGE-L | FID | Semantic | Direction |
| *Clothing* | | | | | | |
| NoAttack | 100 | 100 | 100 | 0 | 0.0306 | 0 |
| VENOMREC | 94.95 | 90.91 | 92.09 | 42.89 | 0.0033 | 0.0459 |
| *Sports* | | | | | | |
| NoAttack | 100 | 100 | 100 | 0 | 0.0178 | 0 |
| VENOMREC | 91.84 | 84.85 | 89.09 | 36.79 | 0.0046 | 0.0841 |
| *Toys* | | | | | | |
| NoAttack | 100 | 100 | 100 | 0 | 0.0282 | 0 |
| VENOMREC | 94.95 | 90.91 | 92.09 | 43.20 | 0.0061 | 0.0003 |

$\{0.05\%, 0.1\%, 0.3\%, 0.5\%, 0.7\%\}$ on the Clothing dataset in Figure 3. As illustrated in the heatmaps (Figure 3), where darker colours represent higher attack/recommendation performance, we observe the following trends:

First, VENOMREC achieves measurable impact even at the lowest poisoning ratio ($\rho = 0.05\%$). At this level, the target item's exposure (ER@5) increases from zero to approximately $0.03$, while recommendation utility metrics (HR and NDCG) remain nearly equivalent to the clean baseline. This suggests that the MLLM's cross-modal fusion is responsive to the synchronised semantic signals introduced by our CIP module, requiring only a marginal fraction of malicious data to initiate the recommendation steering. Second, as the poisoning ratio $\rho$ increases, the attack effectiveness (ER) exhibits a non-linear upward trend, whereas the decline in recommendation utility follows a more gradual trajectory. Specifically, at $\rho = 0.7\%$, ER@5 reaches **0.8543** in the Few-Shot setting and **0.9014** in Zero-Shot. We also observed that the Zero-Shot setting shows a more pronounced response to increasing ratios than the Few-Shot setting. At $\rho = 0.7\%$, the Zero-Shot ER@5 exceeds the Few-Shot result by approximately 5 percentage points. This underscores the finding that without dense historical interaction priors to provide a stabilising context, MLLMs rely more heavily on content-based semantic grounding. Consequently, the model becomes more susceptible to the coordinated perturbations injected via cross-modal interactive poisoning.

Beyond the hyperparameter analysis of compromised user ratio, we further evaluate the scalability of VENOMREC under multi-target scenarios by varying the number of target items $\kappa \in \{1, 3, 5\}$ in Appendix A.5. Results indicate that while ER naturally decreases due to the competition effect among targets, VENOMREC still consistently outperforms Shadowcast by a substantial margin and maintains stable utility (HR/NDCG) across all settings.

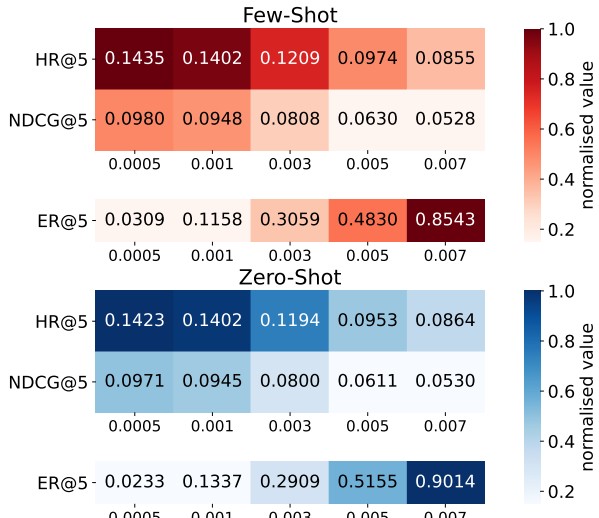

*Figure 3.* Impact of the compromised user ratio $\rho$ on attack effectiveness (ER@5) and recommendation utility (HR@5, NDCG@5) under Few-Shot and Zero-Shot settings on the Clothing dataset.

## 5.4. Ablation Study

To disentangle the contributions of components within VENOMREC, we evaluate four variants on the Clothing dataset (Table 4). Starting from the NoAttack baseline, we cumulatively integrate: (1) **Tab**: Interaction injection (EA); (2) **Img**: Visual perturbation; (3) **Txt**: Textual perturbation; and (4) &: Cross-modal interactive perturbation (CIP).

**Effectiveness of Exposure Alignment (Tab).** The *Tab-only* variant isolates the effect of Exposure Alignment. By injecting adversarial interaction sessions that co-locate the target with high-exposure anchors while keeping the item content unchanged, we observe a non-trivial lift in broad exposure compared to the NoAttack baseline. In the Few-Shot setting, ER@20 rises from 0 to $0.2669$. However, without content-based steering, the target fails to penetrate the top-most ranks, as ER@5 remains negligible at $0.0142$. This confirms that collaborative signals alone provide a necessary "location prior" but are insufficient for precise ranking in content-driven MLLM-RecSys.

**Benefit of Multimodal Perturbation.** We next assess the impact of adding content perturbations. Integrating visual perturbations (*Tab+Img*) yields a moderate gain, with ER@20 improving to $0.4173$. Further adding textual perturbations independently (*Tab+Img+Txt*) raises ER@20 to $0.5617$. This cumulative improvement demonstrates that attacking multiple modalities is more effective than attacking one, as it targets both sensory inputs of the victim model. However, these perturbations remain *independent*, meaning they are not jointly optimised to satisfy cross-modal consistency constraints.

**The Decisive Role of Interaction.** The most critical finding is the substantial performance leap achieved by enabling the interaction loop (*Full* VENOMREC). By synchronising visual and textual edits via attention-guided saliency, ER@20 surges to **0.7170** (Few-Shot) and **0.7317** (Zero-Shot). This margin, representing an absolute increase of approximately 0.15 over the non-interactive variant, empirically validates our core hypothesis: independent perturbations are partially filtered by the model's consensus mechanism, whereas interactive co-adaptation successfully creates a unified, malicious semantic signal that dominates the fusion process.

## 5.5. Robustness Against Defences

We further examine the robustness of VENOMREC under three representative defences, i.e., **AE Filter** (Hao et al., 2019; Hawkins et al., 2002), **LexNorm** (Bitton et al., 2022) and **Freq-Rarity** (Mozes et al., 2021), adapted to our MLLM-RecSys setting, spanning both the interaction and textual channels. The details of these adapted defences are provided in Appendix A.3. Table 5 reports the performance of VENOMREC on the Clothing dataset under each defence in both Few-Shot and Zero-Shot settings.

*Table 4.* Attack performance of VENOMREC on Clothing dataset under Few-Shot and Zero-Shot settings.

| Attack Variants | | | | Few-Shot (Clothing) | | | | | | | | | Zero-Shot (Clothing) | | | | | | | | |
|---|---|---|---|---|---|---|---|---|---|---|---|---|---|---|---|---|---|---|---|---|---|
| | | | | Top-5 | | | Top-10 | | | Top-20 | | | Top-5 | | | Top-10 | | | Top-20 | | |
| Tab | Img | Txt | & | HR | NDCG | ER | HR | NDCG | ER | HR | NDCG | ER | HR | NDCG | ER | HR | NDCG | ER | HR | NDCG | ER |
| − | − | − | − | 0.1439 | 0.0989 | 0 | 0.2225 | 0.1242 | 0 | 0.3421 | 0.1543 | 0 | 0.1437 | 0.0985 | 0 | 0.2218 | 0.1236 | 0 | 0.3386 | 0.1530 | 0 |
| ✓ | − | − | − | 0.1437 | 0.0986 | 0.0142 | 0.2182 | 0.1225 | 0.0327 | 0.3352 | 0.1520 | 0.2669 | 0.1425 | 0.0985 | 0.0145 | 0.2195 | 0.1231 | 0.0346 | 0.3351 | 0.1523 | 0.2904 |
| ✓ | ✓ | − | − | 0.1406 | 0.0962 | 0.0349 | 0.2182 | 0.1210 | 0.0878 | 0.3360 | 0.1506 | 0.4173 | 0.1397 | 0.0947 | 0.0381 | 0.2170 | 0.1194 | 0.0947 | 0.3360 | 0.1494 | 0.4220 |
| ✓ | ✓ | ✓ | − | 0.1442 | 0.0982 | 0.0718 | 0.2233 | 0.1236 | 0.1652 | 0.3381 | 0.1526 | 0.5617 | 0.1443 | 0.0977 | 0.0898 | 0.2237 | 0.1231 | 0.1931 | 0.3377 | 0.1519 | 0.5881 |
| ✓ | ✓ | ✓ | ✓ | 0.1402 | 0.0948 | 0.1158 | 0.2199 | 0.1204 | 0.2596 | 0.3377 | 0.1500 | 0.7170 | 0.1402 | 0.0945 | 0.1337 | 0.2206 | 0.1203 | 0.2820 | 0.3381 | 0.1499 | 0.7317 |

*Table 5.* Robustness of VENOMREC on the Clothing dataset under three representative defences spanning interaction and textual channels.

| Defence | Few-Shot (Clothing) | | | | | | | | | Zero-Shot (Clothing) | | | | | | | | |
|---|---|---|---|---|---|---|---|---|---|---|---|---|---|---|---|---|---|---|
| | Top-5 | | | Top-10 | | | Top-20 | | | Top-5 | | | Top-10 | | | Top-20 | | |
| | HR | NDCG | ER | HR | NDCG | ER | HR | NDCG | ER | HR | NDCG | ER | HR | NDCG | ER | HR | NDCG | ER |
| NoDefence | 0.1402 | 0.0948 | 0.1158 | 0.2199 | 0.1204 | 0.2596 | 0.3377 | 0.1500 | 0.7170 | 0.1402 | 0.0945 | 0.1337 | 0.2206 | 0.1203 | 0.2820 | 0.3381 | 0.1499 | 0.7317 |
| AE Filter | 0.1377 | 0.0926 | 0.1135 | 0.2178 | 0.1182 | 0.2591 | 0.3387 | 0.1487 | 0.7174 | 0.1365 | 0.0918 | 0.1297 | 0.2179 | 0.1179 | 0.2783 | 0.3363 | 0.1477 | 0.7280 |
| LexNorm | 0.1433 | 0.0981 | 0.1482 | 0.2191 | 0.1224 | 0.2998 | 0.3364 | 0.1519 | 0.7100 | 0.1410 | 0.0965 | 0.1361 | 0.2168 | 0.1208 | 0.2755 | 0.3357 | 0.1508 | 0.6743 |
| Freq-Rarity | 0.1385 | 0.0946 | 0.0998 | 0.2168 | 0.1197 | 0.2433 | 0.3354 | 0.1495 | 0.6820 | 0.1361 | 0.0926 | 0.0749 | 0.2162 | 0.1183 | 0.1946 | 0.3351 | 0.1482 | 0.6357 |

Across all defences, ER@20 remains at least $0.6357$, while HR and NDCG remain close to their no-defence values. Thus, the defences fail to eliminate target promotion, and VENOMREC maintains overall recommendation utility close to the no-defence setting. Specifically, the AE Filter changes ER@20 only marginally, from $0.7170$ to $0.7174$ in Few-Shot and from $0.7317$ to $0.7280$ in Zero-Shot. This result is consistent with the scope of the filter: it operates on features derived from user histories and does not directly sanitise modified item content. The textual defences reduce ER@20 to different extents: LexNorm yields $0.7100$ in Few-Shot and $0.6743$ in Zero-Shot, while Freq-Rarity yields $0.6820$ and $0.6357$, respectively, compared with $0.7170$ and $0.7317$ without defence. These results show that the evaluated textual sanitisation baselines alone do not eliminate the promotion effect of VENOMREC.

## 6. Conclusion

In this work, we formalise and investigate the threat of cross-modal interactive poisoning against MLLM-based Recommender Systems. Our study reveals a fundamental security paradox in content-grounded recommendation: while cross-modal consensus typically serves as a natural defence that anchors semantic reasoning against unimodal noise, it simultaneously introduces a critical vulnerability when an adversary can synchronise perturbations across modalities. We propose VENOMREC, a framework that leverages coordinated textual and visual perturbations to establish a stable malicious consensus within the fusion mechanism, thereby exposing this risk. By leveraging Exposure Alignment to identify high-exposure hotspots and Cross-modal Interactive Perturbation to exploit the model's internal attention mechanisms, VENOMREC effectively and stealthily steers the victim model's fused representations without compromising overall recommendation utility. Extensive evaluations on four real-world datasets demonstrate that our approach consistently outperforms existing interaction-level and unimodal baselines, achieving a substantial mean ER@20 of 0.73 and maintaining effectiveness even in challenging Zero-Shot scenarios.

Our findings challenge the prevailing assumption that multimodal integration inherently enhances system robustness. Instead, we demonstrate that the very fusion mechanism designed for resilience can be repurposed to amplify malicious signals. We hope this work serves as a foundation for the community to re-examine the security landscape of MLLM-RecSys under realistic poisoning threats and inspires the development of next-generation defences that account for cross-modal adversarial alignment.

## Acknowledgements

This research is supported by the RIE2025 Industry Alignment Fund – Industry Collaboration Projects (IAF-ICP) (Award I2301E0026), administered by A*STAR, as well as supported by Alibaba Group and NTU Singapore through Alibaba-NTU Global e-Sustainability CorpLab (ANGEL). This research is also supported by the Ministry of Education, Singapore, under its Academic Research Fund Tier 2 (Award MOE-T2EP20125-0005).

## Impact Statement

This paper presents work whose goal is to advance the field of Machine Learning. There are many potential societal consequences of our work, none of which we feel must be specifically highlighted here.

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

# A. Appendix

## A.1. Algorithm

The complete procedure of VENOMREC is summarised in Algorithm 1. The framework operates in two sequential phases to transform clean target items into poisoned samples that are semantically aligned with high-exposure regions.

**Phase I: Target Destination Construction (Lines 3–5).** The process begins by establishing the attack direction via **Exposure Alignment**. The algorithm first mines a set of high-exposure anchor items $\mathcal{H}'$ from the candidate pool $\mathcal{I}$ using observable popularity signals (Line 4). These anchors are then encoded by the proxy model $\phi$ and aggregated to compute the normalised target centroid $\mathbf{z}^\star$ (Line 5). This centroid acts as a stable semantic beacon, guiding the subsequent perturbation process toward a region of the latent space favoured by the recommender system.

**Phase II: Iterative Cross-modal Co-adaptation (Lines 6–24).** For each compromised user $u$ and target item $i^\dagger$, VENOM-REC enters the **Cross-modal Interactive Perturbation** loop. The clean content $(t_{i^\dagger}, v_{i^\dagger})$ is initialized as the starting point $(\tilde{t}, \tilde{v})$. The optimisation proceeds iteratively for up to $R$ rounds:

- **Dynamic Saliency Mining (Lines 12–13):** In each iteration, the algorithm re-evaluates the surrogate model's cross-modal focus. By extracting the cross-modal attention matrix $\mathbf{A}$, it computes updated sensitivity scores $s_\ell$ and $s_p$ to identify the textual tokens and visual patches that currently dominate the fusion process. New binary masks $\mathbf{m}_{\text{txt}}$ and $\mathbf{m}_{\text{vis}}$ are generated to target the top-$k$ most influential elements.
- **Synchronised Perturbation (Lines 14–18):** The visual and textual modalities are updated sequentially but interdependently. The visual features are perturbed along an estimated gradient-free direction $\mathbf{d}_r$, projected onto the $\ell_\infty$ ball to ensure imperceptibility (Line 16). Following this, the textual tokens are optimised by sampling candidates that maximise alignment with $\mathbf{z}^\star$ given the *current* (perturbed) visual state, strictly adhering to the stealthiness constraints $\mathcal{B}$ (Line 18).

The iterative process terminates early if the cosine similarity between the poisoned multimodal embedding $\phi(\tilde{t}, \tilde{v})$ and the target centroid $\mathbf{z}^\star$ exceeds a predefined threshold, ensuring computational efficiency. Finally, the optimised poisoned pairs are aggregated into the malicious dataset $\hat{\mathcal{D}}_{\text{mal}}$.

## A.2. More Description of Experimental Settings

**More Description of Attack Baselines.** We compare VENOMREC against three categories of representative baselines:

- **DirectBoost (Lam & Riedl, 2004):** injects target-only sessions into interaction histories to inflate item frequency.
- **RandomAttack (O'Mahony et al., 2005):** appends the target item at random positions within user sequences.
- **PopularAttack (Zhou et al., 2015):** prepends popular items to induce a distributional shift in the recommendation prior.
- **TextFooler (Jin et al., 2020):** replaces sensitive tokens with synonyms based on importance ranking.
- **DeepwordBug (Gao et al., 2018):** applies character-level transformations to induce out-of-vocabulary errors.
- **PuncAttack (Formento et al., 2023):** inserts subtle punctuation marks to disrupt semantic comprehension.
- **BERTAttack (Li et al., 2020):** leverages masked language models for context-aware adversarial token replacements.
- **INSA (Liu & Larson, 2021):** crafts pixel-level adversarial perturbations on the item image (under an $\ell_\infty$ budget) to maximise the predicted preference score over all user embeddings.
- **EXPA (Liu & Larson, 2021):** crafts pixel-level adversarial perturbations on the item image to pull its visual feature toward a chosen high-exposure hook item in the embedding space.
- **Shadowcast (Xu et al., 2024):** generates adversarial images to perturb multimodal embeddings.

**More Description of Metrics.** To comprehensively evaluate the stealthiness and mechanism of the cross-modal perturbations, we introduce two specific metrics:

- *Semantic Alignment (SA).* This metric evaluates whether the adversarial perturbations disrupt the fundamental semantic consistency between the visual and textual modalities. It is calculated as the cosine similarity between the embeddings of the poisoned image and the poisoned text. By comparing this score against the similarity of the original clean pair, we assess whether the attack preserves the natural correlation expected in benign user-generated content. A stable SA score implies that the modified text still descriptively matches the modified image, ensuring the attack remains imperceptible to consistency-based filters.

**Algorithm 1** The VENOMREC Framework

---

1: **Input:** Target items $\mathcal{I}^\dagger$, compromised users $\mathcal{U}_{\text{mal}}$, proxy model $\phi$, iteration budget $R$, step size $\eta$, stealthiness set $\mathcal{B}$.
2: **Output:** Poisoned malicious dataset $\tilde{\mathcal{D}}_{\text{mal}}$.
3: *// Module I: Exposure Alignment (EA)*
4: Mine high-exposure anchor items $\mathcal{H}' \subseteq \mathcal{I}$ via popularity signals;
5: Compute destination: $\mathbf{z}^\star \leftarrow \text{Norm}\left(\frac{1}{|\mathcal{H}'|}\sum_{j\in\mathcal{H}'}\phi(t_j, v_j)\right)$;
6: *// Module II: Cross-modal Interactive Perturbation (CIP)*
7: $\tilde{\mathcal{D}}_{\text{mal}} \leftarrow \emptyset$;
8: **for all** compromised user $u \in \mathcal{U}_{\text{mal}}$ and target $i^\dagger \in \mathcal{I}^\dagger$ **do**
9:     Initialize $(\tilde{t}, \tilde{v}) \leftarrow (t_{i^\dagger}, v_{i^\dagger})$;
10:     **for** $r = 1$ **to** $R$ **do**
11:         *// Interactive Saliency Mining*
12:         Extract attention $\mathbf{A}$ and compute sensitivity scores $s_\ell, s_p$;
13:         Update masks $\mathbf{m}_{\text{txt}}, \mathbf{m}_{\text{vis}}$ based on top-$k$ sensitivity;
14:         *// Visual Update (Eq. 7)*
15:         Estimate direction $\mathbf{d}_r$ to maximise $\cos(\phi(\tilde{t}, \tilde{v}), \mathbf{z}^\star)$;
16:         $\mathbf{v}_{r+1} \leftarrow \Pi_\epsilon^\infty(\mathbf{v}_r + \eta \cdot \text{sign}(\mathbf{d}_r \odot \mathbf{m}_{\text{vis}}))$;
17:         *// Textual Update*
18:         Sample candidates for $\mathbf{m}_{\text{txt}}$ and select best $\tilde{t}$ s.t. $\mathcal{B}$;
19:         **if** $\cos(\phi(\tilde{t}, \tilde{v}), \mathbf{z}^\star) > $ threshold **then**
20:             **break**;
21:         **end if**
22:     **end for**
23:     $\tilde{\mathcal{D}}_{\text{mal}} \leftarrow \tilde{\mathcal{D}}_{\text{mal}} \cup \{(\tilde{t}, \tilde{v})\}$;
24: **end for**
25: **return** $\tilde{\mathcal{D}}_{\text{mal}}$;

---

- *Latent Direction Alignment (LDA).* This metric measures whether the perturbation vectors of the image and text modalities are oriented in the same direction within the joint embedding space. Specifically, we calculate the cosine similarity between the visual displacement vector (poisoned minus clean) and the textual displacement vector. A high positive LDA indicates that both modalities are collaboratively steering the fused representation toward the target centroid ("lying in the same direction"), whereas a negative or low value suggests that the modalities are sending conflicting signals, which would likely be suppressed by the model's consensus mechanism.

### A.3. More Description of Defence Baselines

We evaluate VENOMREC against three representative defences adapted to our MLLM-RecSys setting:

- **AE Filter (Hao et al., 2019; Hawkins et al., 2002):** trains an autoencoder on clean user-history statistics (length, unique-item ratio, popularity moments, and a tail-popularity statistic) and removes injected user profiles whose reconstruction error exceeds a threshold.

- **LexNorm (Bitton et al., 2022):** applies rule-based token normalisation (lowercasing, punctuation stripping, length capping, deduplication) to the poisoned item text.

- **Freq-Rarity (Mozes et al., 2021):** removes tokens that are absent or of low frequency in the clean review and item-metadata corpus.

### A.4. Attack Effectiveness on T5-base Backbone

To assess the transferability of VENOMREC to larger backbone architectures, we evaluate its performance on a scaled-up victim model based on **T5-base**. Table 6 summarises the results across Clothing, Sports, and Toys datasets under both Few-Shot and Zero-Shot settings.

**Consistent Dominance Across Domains.** VENOMREC consistently achieves the highest targeted exposure rates (ER) across all three datasets. On the Clothing dataset, our method attains an ER@20 of 0.4850 in the Few-Shot setting, outperforming the strongest multimodal baseline, Shadowcast (0.1760), by 0.3090 absolute ER@20 points. This dominance

*Table 6.* Performance comparison of VENOMREC against various baselines with Clothing, Sports, and Toys datasets on T5-base MLLM-RecSys (K=5, 10, and 20) under Few-Shot and Zero-Shot settings.

| | Few-Shot (Clothing) | | | | | | | | | Zero-Shot (Clothing) | | | | | | | | |
| | Top-5 | | | Top-10 | | | Top-20 | | | Top-5 | | | Top-10 | | | Top-20 | | |
| Attack | HR | NDCG | ER | HR | NDCG | ER | HR | NDCG | ER | HR | NDCG | ER | HR | NDCG | ER | HR | NDCG | ER |
|---|---|---|---|---|---|---|---|---|---|---|---|---|---|---|---|---|---|---|
| NoAttack | 0.0866 | 0.0548 | 0 | 0.1602 | 0.0782 | 0 | 0.2792 | 0.1081 | 0 | 0.0862 | 0.0544 | 0 | 0.1598 | 0.0779 | 0 | 0.2798 | 0.1081 | 0 |
| DirectBoost | 0.0848 | 0.0530 | 0.0001 | 0.1582 | 0.0765 | 0.0001 | 0.2808 | 0.1072 | 0.0575 | 0.0849 | 0.0530 | 0.0001 | 0.1582 | 0.0764 | 0.0002 | 0.2809 | 0.1072 | 0.0545 |
| RandomAttack | 0.0851 | 0.0526 | 0.0005 | 0.1560 | 0.0752 | 0.0020 | 0.2816 | 0.1067 | 0.0946 | 0.0851 | 0.0526 | 0.0004 | 0.1559 | 0.0752 | 0.0017 | 0.2811 | 0.1066 | 0.0900 |
| PopularAttack | 0.0860 | 0.0539 | 0 | 0.1599 | 0.0775 | 0.0001 | 0.2806 | 0.1078 | 0.0691 | 0.0846 | 0.0534 | 0 | 0.1605 | 0.0776 | 0.0001 | 0.2813 | 0.1080 | 0.0666 |
| TextFooler | 0.0847 | 0.0532 | 0.0002 | 0.1567 | 0.0762 | 0.0009 | 0.2786 | 0.1068 | 0.1239 | 0.0850 | 0.0533 | 0.0002 | 0.1566 | 0.0761 | 0.0009 | 0.2781 | 0.1066 | 0.1263 |
| DeepwordBug | 0.0834 | 0.0524 | 0.0003 | 0.1567 | 0.0758 | 0.0024 | 0.2769 | 0.1059 | 0.1815 | 0.0834 | 0.0525 | 0.0003 | 0.1568 | 0.0758 | 0.0024 | 0.2780 | 0.1063 | 0.1807 |
| PuncAttack | 0.0847 | 0.0529 | 0.0001 | 0.1566 | 0.0758 | 0.0007 | 0.2809 | 0.1070 | 0.0849 | 0.0852 | 0.0528 | 0.0001 | 0.1547 | 0.0750 | 0.0007 | 0.2811 | 0.1068 | 0.0823 |
| BERTAttack | 0.0846 | 0.0532 | 0.0040 | 0.1570 | 0.0764 | 0.0155 | 0.2804 | 0.1074 | 0.2858 | 0.0849 | 0.0531 | 0.0033 | 0.1559 | 0.0758 | 0.0146 | 0.2790 | 0.1067 | 0.2821 |
| INSA | 0.0923 | 0.0591 | 0.0020 | 0.1631 | 0.0817 | 0.0092 | 0.2843 | 0.1122 | 0.2565 | 0.0910 | 0.0578 | 0.0014 | 0.1622 | 0.0805 | 0.0065 | 0.2855 | 0.1115 | 0.1861 |
| EXPA | 0.0852 | 0.0537 | 0.0022 | 0.1594 | 0.0774 | 0.0096 | 0.2808 | 0.1079 | 0.2613 | 0.0847 | 0.0532 | 0.0018 | 0.1593 | 0.0771 | 0.0080 | 0.2811 | 0.1077 | 0.2497 |
| Shadowcast | 0.0833 | 0.0526 | 0.0004 | 0.1553 | 0.0756 | 0.0019 | 0.2772 | 0.1062 | 0.1760 | 0.0836 | 0.0525 | 0.0004 | 0.1552 | 0.0754 | 0.0018 | 0.2769 | 0.1059 | 0.1716 |
| **VENOMREC** | 0.0806 | 0.0500 | 0.0237 | 0.1532 | 0.0732 | 0.0694 | 0.2756 | 0.1039 | 0.4850 | 0.0799 | 0.0497 | 0.0210 | 0.1529 | 0.0731 | 0.0630 | 0.2763 | 0.1041 | 0.4676 |

| | Few-Shot (Sports) | | | | | | | | | Zero-Shot (Sports) | | | | | | | | |
| | Top-5 | | | Top-10 | | | Top-20 | | | Top-5 | | | Top-10 | | | Top-20 | | |
| Attack | HR | NDCG | ER | HR | NDCG | ER | HR | NDCG | ER | HR | NDCG | ER | HR | NDCG | ER | HR | NDCG | ER |
|---|---|---|---|---|---|---|---|---|---|---|---|---|---|---|---|---|---|---|
| NoAttack | 0.1176 | 0.0785 | 0 | 0.1950 | 0.1033 | 0 | 0.3153 | 0.1335 | 0 | 0.1180 | 0.0785 | 0 | 0.1952 | 0.1032 | 0 | 0.3162 | 0.1336 | 0 |
| DirectBoost | 0.1183 | 0.0794 | 0 | 0.1982 | 0.1050 | 0.0007 | 0.3172 | 0.1349 | 0.1066 | 0.1185 | 0.0796 | 0 | 0.1972 | 0.1047 | 0.0008 | 0.3175 | 0.1351 | 0.1120 |
| RandomAttack | 0.1189 | 0.0793 | 0.0001 | 0.1951 | 0.1038 | 0.0011 | 0.3189 | 0.1349 | 0.1491 | 0.1185 | 0.0793 | 0.0001 | 0.1966 | 0.1043 | 0.0012 | 0.3192 | 0.1351 | 0.1481 |
| PopularAttack | 0.1121 | 0.0735 | 0.0014 | 0.1922 | 0.0992 | 0.0059 | 0.3086 | 0.1284 | 0.2621 | 0.1129 | 0.0739 | 0.0016 | 0.1919 | 0.0992 | 0.0062 | 0.3101 | 0.1289 | 0.2609 |
| TextFooler | 0.1156 | 0.0770 | 0.0010 | 0.1929 | 0.1017 | 0.0052 | 0.3156 | 0.1326 | 0.2726 | 0.1156 | 0.0770 | 0.0010 | 0.1929 | 0.1017 | 0.0052 | 0.3156 | 0.1326 | 0.2726 |
| DeepwordBug | 0.1150 | 0.0774 | 0.0031 | 0.1932 | 0.1024 | 0.0146 | 0.3161 | 0.1334 | 0.3532 | 0.1152 | 0.0772 | 0.0032 | 0.1937 | 0.1023 | 0.0129 | 0.3170 | 0.1333 | 0.3449 |
| PuncAttack | 0.1168 | 0.0777 | 0.0016 | 0.1963 | 0.1032 | 0.0062 | 0.3166 | 0.1335 | 0.2475 | 0.1165 | 0.0775 | 0.0016 | 0.1955 | 0.1028 | 0.0066 | 0.3166 | 0.1333 | 0.2531 |
| BERTAttack | 0.1169 | 0.0781 | 0.0005 | 0.1955 | 0.1032 | 0.0022 | 0.3179 | 0.1340 | 0.1984 | 0.1167 | 0.0779 | 0.0006 | 0.1954 | 0.1030 | 0.0025 | 0.3178 | 0.1338 | 0.1998 |
| INSA | 0.1196 | 0.0801 | 0.0005 | 0.1987 | 0.1054 | 0.0022 | 0.3203 | 0.1360 | 0.1521 | 0.1192 | 0.0801 | 0.0005 | 0.1983 | 0.1053 | 0.0024 | 0.3193 | 0.1359 | 0.1519 |
| EXPA | 0.1179 | 0.0783 | 0.0001 | 0.1977 | 0.1039 | 0.0004 | 0.3185 | 0.1343 | 0.0880 | 0.1173 | 0.0780 | 0.0001 | 0.1972 | 0.1036 | 0.0004 | 0.3179 | 0.1340 | 0.0909 |
| Shadowcast | 0.1154 | 0.0771 | 0.0088 | 0.1921 | 0.1016 | 0.0287 | 0.3146 | 0.1324 | 0.3823 | 0.1153 | 0.0775 | 0.0087 | 0.1916 | 0.1019 | 0.0285 | 0.3150 | 0.1330 | 0.3842 |
| **VENOMREC** | 0.1175 | 0.0787 | 0.0252 | 0.1980 | 0.1045 | 0.0797 | 0.3168 | 0.1343 | 0.5580 | 0.1169 | 0.0782 | 0.0216 | 0.1974 | 0.1039 | 0.0707 | 0.3165 | 0.1339 | 0.5398 |

| | Few-Shot (Toys) | | | | | | | | | Zero-Shot (Toys) | | | | | | | | |
| | Top-5 | | | Top-10 | | | Top-20 | | | Top-5 | | | Top-10 | | | Top-20 | | |
| Attack | HR | NDCG | ER | HR | NDCG | ER | HR | NDCG | ER | HR | NDCG | ER | HR | NDCG | ER | HR | NDCG | ER |
|---|---|---|---|---|---|---|---|---|---|---|---|---|---|---|---|---|---|---|
| NoAttack | 0.0992 | 0.0632 | 0 | 0.1728 | 0.0868 | 0 | 0.2943 | 0.1173 | 0 | 0.1005 | 0.0638 | 0 | 0.1728 | 0.0870 | 0 | 0.2953 | 0.1178 | 0 |
| DirectBoost | 0.0969 | 0.0624 | 0.0980 | 0.1665 | 0.0846 | 0.2817 | 0.2874 | 0.1149 | 0.7579 | 0.0971 | 0.0622 | 0.0895 | 0.1652 | 0.0840 | 0.2633 | 0.2893 | 0.1151 | 0.7540 |
| RandomAttack | 0.0934 | 0.0604 | 0.4992 | 0.1660 | 0.0836 | 0.7391 | 0.2913 | 0.1150 | 0.9543 | 0.0943 | 0.0607 | 0.4800 | 0.1669 | 0.0839 | 0.7223 | 0.2896 | 0.1147 | 0.9495 |
| PopularAttack | 0.0913 | 0.0582 | 0.0518 | 0.1611 | 0.0806 | 0.1501 | 0.2861 | 0.1120 | 0.5586 | 0.0915 | 0.0582 | 0.0446 | 0.1621 | 0.0808 | 0.1351 | 0.2871 | 0.1121 | 0.5334 |
| TextFooler | 0.0981 | 0.0631 | 0.4412 | 0.1732 | 0.0872 | 0.6404 | 0.2954 | 0.1178 | 0.9180 | 0.0979 | 0.0629 | 0.4462 | 0.1744 | 0.0874 | 0.6457 | 0.2940 | 0.1173 | 0.9191 |
| DeepwordBug | 0.0954 | 0.0612 | 0.1556 | 0.1709 | 0.0853 | 0.3245 | 0.2966 | 0.1169 | 0.7349 | 0.0957 | 0.0616 | 0.1532 | 0.1716 | 0.0858 | 0.3257 | 0.2975 | 0.1174 | 0.7336 |
| PuncAttack | 0.0971 | 0.0616 | 0.2030 | 0.1710 | 0.0852 | 0.3634 | 0.2958 | 0.1164 | 0.7400 | 0.0974 | 0.0625 | 0.3047 | 0.1728 | 0.0866 | 0.4994 | 0.2940 | 0.1170 | 0.8549 |
| BERTAttack | 0.1000 | 0.0639 | 0.2400 | 0.1761 | 0.0883 | 0.3980 | 0.2963 | 0.1185 | 0.7455 | 0.1014 | 0.0643 | 0.2322 | 0.1764 | 0.0883 | 0.3887 | 0.2976 | 0.1187 | 0.7343 |
| INSA | 0.0963 | 0.0621 | 0.3084 | 0.1687 | 0.0853 | 0.5095 | 0.2905 | 0.1158 | 0.8419 | 0.0960 | 0.0619 | 0.2984 | 0.1677 | 0.0848 | 0.5004 | 0.2913 | 0.1158 | 0.8337 |
| EXPA | 0.0945 | 0.0607 | 0.4321 | 0.1689 | 0.0844 | 0.6096 | 0.2910 | 0.1150 | 0.8859 | 0.0936 | 0.0604 | 0.4241 | 0.1679 | 0.0842 | 0.6039 | 0.2916 | 0.1152 | 0.8807 |
| Shadowcast | 0.1001 | 0.0639 | 0.3161 | 0.1741 | 0.0876 | 0.5257 | 0.2960 | 0.1181 | 0.8643 | 0.1012 | 0.0641 | 0.3122 | 0.1745 | 0.0875 | 0.5175 | 0.2956 | 0.1179 | 0.8619 |
| **VENOMREC** | 0.0885 | 0.0557 | 0.6578 | 0.1594 | 0.0784 | 0.8129 | 0.2835 | 0.1095 | 0.9646 | 0.0886 | 0.0558 | 0.6574 | 0.1602 | 0.0787 | 0.8137 | 0.2852 | 0.1100 | 0.9639 |

is even more pronounced on the **Toys** dataset, where VENOMREC achieves near-saturation with an ER@20 of **0.9646** (Few-Shot) and **0.9639** (Zero-Shot), effectively monopolising the recommendation lists for the target items. These results confirm that the cross-modal "hotspots" identified by our Exposure Alignment module are not artefacts of a specific model size but represent robust semantic vulnerabilities that persist in larger parameter spaces.

**Robustness in Zero-Shot Scenarios.** The attack demonstrates remarkable stability when transitioning to the Zero-Shot setting. For instance, on the **Sports** dataset, the ER@20 remains virtually unchanged, shifting from 0.5580 (Few-Shot) to **0.5398** (Zero-Shot). This suggests that VENOMREC does not rely on overfitting to specific prompt templates or Few-Shot examples. Instead, it successfully embeds the malicious signal into the intrinsic semantic alignment between the visual and textual modalities, allowing the attack to succeed even when explicit interaction examples are absent during inference.

**Recommendation Utility.** Crucially, this aggressive promotion does not come at the expense of general user experience. Across all datasets, the Hit Ratio (HR) and NDCG scores of the victim model under VENOMREC remain comparable to the clean baseline (NoAttack). For example, on **Clothing** (Zero-Shot), the HR@20 decreases only marginally from 0.2798 (NoAttack) to 0.2763 (VENOMREC), a negligible drop given the massive increase in target item visibility. This confirms that VENOMREC operates in a "surgical" manner, precisely steering target items without disrupting the global ranking distribution for benign items.

## A.5. Impact of the Number of Target Items

To evaluate the scalability of VENOMREC under more demanding attack scenarios, we investigate the impact of the number of simultaneously targeted items, denoted by $\kappa$. We vary $\kappa \in \{1, 3, 5\}$ on the *Clothing* dataset and measure the trade-off between attack effectiveness and recommendation utility in Table 7.

*Table 7.* Attack performance on the Clothing dataset under different numbers of targeted items.

| Attack | Setting | Top-5 | | | Top-10 | | | Top-20 | | |
|---|---|---|---|---|---|---|---|---|---|---|
| | | HR | NDCG | ER | HR | NDCG | ER | HR | NDCG | ER |
| *κ=1* | | | | | | | | | | |
| Shadowcast | Few-Shot | 0.1404 | 0.0962 | 0.0041 | 0.2156 | 0.1203 | 0.0073 | 0.3319 | 0.1496 | 0.0472 |
| | Zero-Shot | 0.1405 | 0.0957 | 0.0032 | 0.2152 | 0.1196 | 0.0066 | 0.3322 | 0.1491 | 0.0451 |
| VENOMREC | Few-Shot | 0.1402 | 0.0948 | 0.1158 | 0.2199 | 0.1204 | 0.2596 | 0.3377 | 0.1500 | 0.7170 |
| | Zero-Shot | 0.1402 | 0.0945 | 0.1337 | 0.2206 | 0.1203 | 0.2820 | 0.3381 | 0.1499 | 0.7317 |
| *κ=3* | | | | | | | | | | |
| Shadowcast | Few-Shot | 0.1469 | 0.1006 | 0.0021 | 0.2223 | 0.1247 | 0.0038 | 0.3362 | 0.1534 | 0.0414 |
| | Zero-Shot | 0.1444 | 0.0988 | 0.0031 | 0.2207 | 0.1233 | 0.0054 | 0.3363 | 0.1524 | 0.0386 |
| VENOMREC | Few-Shot | 0.1467 | 0.1023 | 0.0152 | 0.2201 | 0.1258 | 0.0365 | 0.3364 | 0.1551 | 0.2904 |
| | Zero-Shot | 0.1466 | 0.1007 | 0.0196 | 0.2202 | 0.1243 | 0.0444 | 0.3360 | 0.1534 | 0.3327 |
| *κ=5* | | | | | | | | | | |
| Shadowcast | Few-Shot | 0.1447 | 0.0990 | 0.0026 | 0.2220 | 0.1238 | 0.0048 | 0.3368 | 0.1527 | 0.0349 |
| | Zero-Shot | 0.1426 | 0.0972 | 0.0025 | 0.2220 | 0.1227 | 0.0046 | 0.3359 | 0.1514 | 0.0306 |
| VENOMREC | Few-Shot | 0.1403 | 0.0957 | 0.0055 | 0.2184 | 0.1207 | 0.0104 | 0.3338 | 0.1497 | 0.1303 |
| | Zero-Shot | 0.1402 | 0.0957 | 0.0059 | 0.2179 | 0.1206 | 0.0111 | 0.3322 | 0.1494 | 0.1380 |

**Performance Dilution with Increased Targets.** As expected, we observe a trade-off between the number of targets and the exposure rate per target. When the attack focus is singular ($\kappa = 1$), VENOMREC achieves a dominant ER@20 of **0.7170** (Few-Shot). As $\kappa$ increases to 3 and 5, the ER@20 naturally decreases to **0.2904** and **0.1303**, respectively. This phenomenon can be attributed to the "competition effect": multiple target items must compete not only with organic items but also against each other for the limited slots in the top-$K$ recommendation lists. Furthermore, the fixed poisoning budget is effectively diluted across multiple semantic directions, weakening the steering signal for any individual item.

**Superior Multi-Target Robustness.** Despite this dilution, VENOMREC exhibits significantly higher robustness than the baseline. At $\kappa = 5$, *Shadowcast* fails almost completely, with an ER@20 of roughly **0.03**, which is statistically indistinguishable from benign exposure. In contrast, VENOMREC maintains a substantial influence (ER@20 $\approx$ **0.13**), demonstrating that our cross-modal interactive perturbation generates a sufficiently potent semantic signal to support multi-target promotion, whereas independent perturbations (as in Shadowcast) struggle to steer the model toward multiple disparate latent regions simultaneously.

**Utility Stability.** Crucially, the global recommendation quality remains stable even as the attack complexity increases. The Hit Ratio (HR@20) for VENOMREC fluctuates only marginally (from $\approx 0.338$ at $\kappa = 1$ to $\approx 0.334$ at $\kappa = 5$). This indicates that VENOMREC promotes multiple targets by displacing "weak" organic recommendations rather than disrupting the core user preference modelling, preserving the stealthiness of the attack even in multi-target scenarios.

## A.6. Transferability to Traditional Multimodal Recommenders

To assess whether the vulnerability exposed by VENOMREC is specific to the VIP5/T5-base MLLM family or extends to a fundamentally different multimodal recommender architecture, we additionally evaluate VENOMREC on **DiffMM**, a traditional multimodal recommender beyond the VIP5/T5-base MLLM family. We use the same Clothing, Sports, and Toys datasets and the same target-item construction protocol as in our main experiments, and compare against *NoAttack* (clean) and *Shadowcast* as the strongest multimodal baseline (cf. Tables 2 and 6). The results are reported in Table 8.

Across all three datasets, VENOMREC remains the dominant attack against DiffMM. On the **Sports** dataset, VENOM-REC achieves **ER@10 = 0.6667**, outperforming Shadowcast by **0.507** absolute ER@10, while maintaining comparable

recommendation utility (e.g., NDCG@10 = 0.1639 versus 0.1635 for NoAttack). Similar trends are observed on **Clothing** (ER@20: 0.5546 → 0.6341) and **Toys** (ER@20: 0.1555 → 0.8963), where VENOMREC consistently improves over Shadowcast while keeping HR and NDCG close to the NoAttack baseline. These results suggest that the vulnerability exploited by VENOMREC is not confined to one specific multimodal recommender implementation, but is more broadly related to multimodal representation fusion and coordinated cross-modal steering.

*Table 8.* Attack performance against **DiffMM** (Jiang et al., 2024), a traditional multimodal recommender beyond the VIP5/T5-base MLLM family, on Clothing, Sports, and Toys. We focus on *Shadowcast* as the strongest multimodal baseline (cf. Tables 2 and 6).

| Dataset | Attack | Top-5 | | | Top-10 | | | Top-20 | | |
|---|---|---|---|---|---|---|---|---|---|---|
| | | HR | NDCG | ER | HR | NDCG | ER | HR | NDCG | ER |
| *Dataset: Sports* | | | | | | | | | | |
| Sports | NoAttack | 0.1919 | 0.1315 | 0 | 0.2916 | 0.1635 | 0 | 0.4320 | 0.1988 | 0 |
| | Shadowcast | 0.1919 | 0.1315 | 0.0810 | 0.2916 | 0.1634 | 0.1597 | 0.4320 | 0.1988 | 0.3109 |
| | **VENOMREC** | 0.1921 | 0.1315 | **0.4416** | 0.2933 | 0.1639 | **0.6667** | 0.4358 | 0.1998 | **0.8870** |
| *Dataset: Clothing* | | | | | | | | | | |
| Clothing | NoAttack | 0.2554 | 0.1893 | 0 | 0.3458 | 0.2184 | 0 | 0.4738 | 0.2506 | 0 |
| | Shadowcast | 0.2553 | 0.1893 | 0.2171 | 0.3458 | 0.2184 | 0.3506 | 0.4739 | 0.2506 | 0.5546 |
| | **VENOMREC** | 0.2551 | 0.1893 | **0.2225** | 0.3456 | 0.2184 | **0.3662** | 0.4731 | 0.2505 | **0.6341** |
| *Dataset: Toys* | | | | | | | | | | |
| Toys | NoAttack | 0.2744 | 0.2078 | 0 | 0.3587 | 0.2349 | 0 | 0.4836 | 0.2663 | 0 |
| | Shadowcast | 0.2743 | 0.2077 | 0.0286 | 0.3586 | 0.2348 | 0.0676 | 0.4837 | 0.2663 | 0.1555 |
| | **VENOMREC** | 0.2736 | 0.2064 | **0.5093** | 0.3564 | 0.2330 | **0.7140** | 0.4818 | 0.2645 | **0.8963** |

## A.7. Transferability to Short-Video Recommendation Domain

To complement the Amazon e-commerce datasets used in our main experiments, we further evaluate VENOMREC on **MicroLens** (Ni et al., 2025), a short-video recommendation dataset with 100,000 users, 19,738 items, and 719,405 interactions. This broadens the empirical coverage from e-commerce products to short-video content with substantially different visual and textual characteristics. We follow the same target-item construction protocol as in the main experiments and compare VENOMREC against *Shadowcast* as the strongest multimodal baseline. The results are reported in Table 9.

On MicroLens, VENOMREC remains highly effective under both Zero-Shot and Few-Shot settings while preserving recommendation utility close to the clean model. Specifically, in Zero-Shot, VENOMREC improves ER@10 from 0.0005 under Shadowcast to **0.7909**; in Few-Shot, it improves ER@10 from 0.0007 to **0.8066**. At the same time, the ranking utility remains nearly unchanged (e.g., HR@10 = 0.2161/0.2173 for Zero-Shot/Few-Shot), staying very close to the clean baseline (HR@10 = 0.2164/0.2163). These results indicate that the vulnerability studied in this paper is not limited to Amazon product datasets, and can also be observed in another domain with markedly different content characteristics.

*Table 9.* Attack performance on the **MicroLens** short-video recommendation dataset under Zero-Shot and Few-Shot settings. We focus on *Shadowcast* as the strongest multimodal baseline (cf. Tables 2 and 6).

| Setting | Attack | Top-5 | | | Top-10 | | | Top-20 | | |
|---|---|---|---|---|---|---|---|---|---|---|
| | | HR | NDCG | ER | HR | NDCG | ER | HR | NDCG | ER |
| Zero-Shot | NoAttack | 0.1219 | 0.0757 | 0 | 0.2164 | 0.1059 | 0 | 0.3637 | 0.1431 | 0 |
| | Shadowcast | 0.1220 | 0.0758 | 0 | 0.2168 | 0.1062 | 0.0005 | 0.3665 | 0.1439 | 0.0314 |
| | **VENOMREC** | 0.1217 | 0.0754 | **0.4158** | 0.2161 | 0.1056 | **0.7909** | 0.3650 | 0.1432 | **0.9864** |
| Few-Shot | NoAttack | 0.1230 | 0.0765 | 0 | 0.2163 | 0.1064 | 0 | 0.3637 | 0.1436 | 0 |
| | Shadowcast | 0.1231 | 0.0761 | 0 | 0.2169 | 0.1061 | 0.0007 | 0.3658 | 0.1436 | 0.0366 |
| | **VENOMREC** | 0.1227 | 0.0758 | **0.4357** | 0.2173 | 0.1061 | **0.8066** | 0.3662 | 0.1436 | **0.9885** |

