# OpenReview forum: "VENOMREC: Cross-Modal Interactive Poisoning for Targeted Promotion in Multimodal LLM Recommender Systems"
_ICML.cc/2026/Conference — ICML 2026 regular_

### Official Review · Reviewer_7JqE · 2026-03-06

**Soundness:** 2
**Presentation:** 2
**Significance:** 2
**Originality:** 3
**Overall Recommendation:** 4
**Confidence:** 3

**Summary:**

This paper focuses on the security issues of the Multimodal Large Language Model Recommendation System (MLLM-RecSys), pointing out that while cross-modal fusion can resist traditional single-modal or interaction-layer poisoning attacks, a new vulnerability exists in the consensus mechanism—synchronous cross-modal poisoning can amplify malicious signals. To address this, the paper proposes a cross-modal interactive poisoning attack framework, VENOMREC. This framework uses Exposure Alignment (EA) to locate high-exposure semantic regions and Cross-Modal Interactive Perturbation (CIP) to generate covert and collaborative text/visual poisoning data, achieving targeted improvement in the exposure of target items without significantly affecting recommendation effectiveness.

**Compliance With Llm Reviewing Policy:**

Affirmed.

**Final Justification:**

The author add additional experiments regarding the traditional MMRec methods and show the effectiveness of the proposed method. Moreover, a new benchmark is evaluated. If the authors include these new results in the revised draft, I think it is worth a positive score.

**Key Questions For Authors:**

1. Can this method be applied to traditional multimodal recommendation models?

**Limitations:**

yes

**Strengths And Weaknesses:**

Strengths:

1. This work formally defines and systematically studies cross-modal interactive poisoning attacks against MLLM-RecSys for the first time, revealing the security paradox of cross-modal fusion.

2. The VENOMREC framework is proposed, which innovatively utilizes cross-modal attention mechanisms to optimize cooperative perturbations while simultaneously satisfying the concealment requirements of single-modal naturalness and cross-modal consistency.

3. The effectiveness, concealment, and robustness of the framework were verified through experiments.

Weakness:

1. The effectiveness of this attack method is closely related to the backbone architecture of the victim model, and its performance drops significantly on larger-scale models. Its reliance on precise utilization of the model's cross-modal attention mechanism reduces its generalization ability.

2. The authors did not specifically discuss "adaptation to different multimodal recommendation models," but they also did not cover non-LM-based multimodal recommendation models or customized fusion architectures.

3. The dataset used is limited to the Amazon e-commerce platform, and datasets from other scenarios (such as short videos, news, etc.) are lacking.

---

> ### Author Rebuttal · Authors · 2026-03-31
>
> We appreciate your recognition of the paper’s novelty and of the central role of cross-modal fusion in the threat. Below we clarify the key points and provide additional evidence where requested.
>
> ### **W.1: Effectiveness Across Backbone Scales.**
> We agree that the attack strength decreases as the backbone scales up. As shown in Table 1 and Table 5, On the Clothing dataset, VENOMREC’s ER20 drops from 0.7170/0.7317 on **T5-small** to 0.4850/0.4676 on **T5-base**, which suggests that larger MLLM backbones are harder to steer. A plausible reason is that **larger models exhibit stronger semantic redundancy and cross-modal consensus**, making weak or independently optimized perturbations easier to absorb. Importantly, however, VENOMREC still substantially outperforms ShadowCast on T5-base (0.1760/0.1716 → 0.4850/0.4676) while preserving utility close to the clean model. These results show that VENOMREC remains effective beyond smaller backbones and continues to **provide clear exposure gains on stronger models**.
>
> ### **W.2/Q1: Applicability to Traditional Multimodal Recommenders.**
> We additionally evaluated VENOMREC on **DiffMM [1], a traditional multimodal recommender** beyond the VIP5-style MLLMs to address your concern. Across the three datasets, VENOMREC achieves an average ER20 of 0.8058, improving over ShadowCast by 0.4654 absolute points on average, while keeping utility essentially unchanged. The gains are strongest on Sports (0.3109 → 0.8870) and Toys (0.1555 → 0.8963), and more moderate on Clothing (0.5546 → 0.6341). **These results show that VENOMREC can be effectively instantiated on a traditional multimodal recommender**.
>
> More importantly, this also helps clarify the role of cross-modal attention in our method. In the VIP5 instantiation, **cross-modal attention** rollout is the guidance-extraction mechanism we use to identify salient token–patch interactions, but it is **not a strict prerequisite of the attack principle itself**.
>
> * Sports dataset
>
> |Attack|HR5|NDCG5|ER5|HR10|NDCG10|ER10|HR20|NDCG20|ER20|
> |---|---|---|---|---|---|---|---|---|---|
> |NoAttack|0.1919|0.1315|0|0.2916|0.1635|0|0.4320|0.1988|0
> |ShadowCast|0.1919|0.1315|0.0810|0.2916|0.1634|0.1597|0.4320|0.1988|0.3109
> |VENOMREC|0.1921|0.1315|0.4416|0.2933|0.1639|0.6667|0.4358|0.1998|0.8870
>
> * Clothing dataset
>
> |Attack|HR5|NDCG5|ER5|HR10|NDCG10|ER10|HR20|NDCG20|ER20|
> |---|---|---|---|---|---|---|---|---|---|
> |NoAttack|0.2554|0.1893|0|0.3458|0.2184|0|0.4738|0.2506|0
> |ShadowCast|0.2553|0.1893|0.2171|0.3458|0.2184|0.3506|0.4739|0.2506|0.5546
> |VENOMREC|0.2551|0.1893|0.2225|0.3456|0.2184|0.3662|0.4731|0.2505|0.6341
>
> * Toys dataset
>
> |Attack|HR5|NDCG5|ER5|HR10|NDCG10|ER10|HR20|NDCG20|ER20|
> |---|---|---|---|---|---|---|---|---|---|
> |NoAttack|0.2744|0.2078|0|0.3587|0.2349|0|0.4836|0.2663|0
> |ShadowCast|0.2743|0.2077|0.0286|0.3586|0.2348|0.0676|0.4837|0.2663|0.1555
> |VENOMREC|0.2736|0.2064|0.5093|0.3564|0.2330|0.7140|0.4818|0.2645|0.8963
>
>
> ### **W.3: More Datasets Evaluation.**
> To complement Amazon datasets (i.e., Clothing, Sports, and Toys) used in our paper, we further evaluated VENOMREC on **MicroLens [2]**, **a short-video recommendation dataset** with **100,000 users, 19,738 items, and 719,405 interactions**. This addition **broadens the empirical coverage** from e-commerce products to short-video content.
>
> **On the MicroLens dataset, VENOMREC remains highly effective in both Zero-Shot (ZS) and Few-Shot (FS) settings while preserving recommendation utility close to NoAttack**. Specifically, in ZS, VENOMREC improves ER10 from 0.0005 under ShadowCast to 0.7909; in FS, it improves ER10 from 0.0007 to 0.8066. At the same time, the ranking utility remains nearly unchanged (e.g., HR10 = 0.2161/0.2173 for ZS/FS), staying very close to the clean model.These results indicate that the observed vulnerability studied in this paper is **not limited to the Amazon product datasets**, and can also be observed in another domain with different content characteristics.
>
> |Attack(ZS)|HR5|NDCG5|ER5|HR10|NDCG10|ER10|HR20|NDCG20|ER20|
> |---|---|---|---|---|---|---|---|---|---|
> |NoAttack|0.1219|0.0757|0|0.2164|0.1059|0|0.3637|0.1431|0|
> |ShadowCast|0.1220|0.0758|0|0.2168|0.1062|0.0005|0.3665|0.1439|0.0314|
> |VENOMREC|0.1217|0.0754|0.4158|0.2161|0.1056|0.7909|0.3650|0.1432|0.9864|
>
> |Attack(FS)|HR5|NDCG5|ER5|HR10|NDCG10|ER10|HR20|NDCG20|ER20|
> |---|---|---|---|---|---|---|---|---|---|
> |NoAttack|0.1230|0.0765|0|0.2163|0.1064|0|0.3637|0.1436|0|
> |ShadowCast|0.1231|0.0761|0|0.2169|0.1061|0.0007|0.3658|0.1436|0.0366|
> |VENOMREC|0.1227|0.0758|0.4357|0.2173|0.1061|0.8066|0.3662|0.1436|0.9885|
>
> ### **Reference**:
> [1] DiffMM: Multi-Modal Diffusion Model for Recommendation, ACM MM 2024.
>
> [2] MicroLens url: https://github.com/westlake-repl/MicroLens/
>
> ---
> **We hope these clarifications and additional results address the your concerns.**

---

> > ### Author Rebuttal · Reviewer_7JqE · 2026-04-03
> >
> > This rebuttal addresses my concerns. I will raise my score. The author should add the related results mentioned in the rebuttal to the revised draft.

---

> > > ### Author Response · Authors · 2026-04-05
> > >
> > > Thank you very much for your thoughtful follow-up and for revisiting our rebuttal so carefully. We truly appreciate your positive reassessment (rating to 4) of the paper after reading the rebuttal, and we are encouraged that the additional results and clarifications were recognized.
> > > We will incorporate these added results and clarifications into the final manuscript.
> > > Thank you again for your careful reading and constructive feedback.

---

### Official Review · Reviewer_yGxZ · 2026-03-09

**Soundness:** 3
**Presentation:** 3
**Significance:** 3
**Originality:** 3
**Overall Recommendation:** 4
**Confidence:** 3

**Summary:**

This paper studies targeted poisoning attacks on multimodal LLM-based recommender systems. It proposes VENOMREC, a two-stage framework consisting of Exposure Alignment (EA), which moves target items toward high-exposure regions in the joint embedding space using popular anchors, and Cross-modal Interactive Perturbation (CIP), which iteratively perturbs salient text tokens and image patches using cross-modal attention. Experiments on three Amazon multimodal recommendation datasets show that VENOMREC significantly increases target exposure while maintaining overall recommendation quality.

**Compliance With Llm Reviewing Policy:**

Affirmed.

**Final Justification:**

This rebuttle address my concerns regarding generalization and stealthiness. I have no further questions and will maintain my score.

**Key Questions For Authors:**

- How sensitive is VENOMREC to the choice of proxy encoder used to approximate the victim model?
- Can the authors provide clearer implementation details for the CIP optimization (text candidate generation, image direction estimation, key hyperparameters)?
- Does the attack generalize to other multimodal recommender architectures beyond VIP5?
- Can stronger content-aware defenses be evaluated?

**Limitations:**

No. The paper should more clearly discuss deployment realism, misuse risks, and safeguards when releasing code or datasets for this attack.

**Strengths And Weaknesses:**

Strengths
- Addresses an important and timely problem: security of multimodal LLM recommender systems.
- Clear attack formulation and intuitive two-stage framework (EA + CIP).
- Strong empirical results with substantial improvements in target exposure.
- Includes useful studies such as ablation, poisoning-budget analysis, and transfer to a larger backbone.

Weaknesses
- Evaluation is limited to one model family (VIP5 with T5 variants) and Amazon datasets, which weakens claims about generality.
- Some implementation details of the attack (e.g., textual candidate generation, gradient-free image updates, hyperparameters) are not fully specified, affecting reproducibility.
- Stealthiness evaluation relies mainly on automatic metrics (ROUGE, FID) without human or moderation-based evaluation.
- Defense evaluation is limited (only one filtering method), and the discussion of societal risks is insufficient for an attack paper.

---

> ### Author Rebuttal · Authors · 2026-03-31
>
> We thank the reviewer for thoughtful feedback and encouraging assessment of the paper’s formulation, novelty, and empirical promise. Below, we address the specific concerns.
>
> ### **W.1/Q.3: Generalization Across Diverse Architectures and Datasets.**
> To address generality concerns, we expanded the evaluation along both axes raised in your review.
>
> **Architecture**. Beyond VIP5, we additionally evaluated VENOMREC on **DiffMM[1]**, a multi-modal diffusion recommender with a substantially different architecture. VENOMREC **consistently outperform ShadowCast on all three datasets**, achieving an average ER20 of 0.8058 with an average gain of 0.4654.`see W.2 of Reviewer jiBT`
>
> **Dataset**. Beyond the Amazon datasets, we further evaluated VENOMREC on **MicroLens[2]**, a short-video recommendation context with distinct modal distributions and user behaviors. VENOMREC **remains highly effective in both Zero-Shot and Few-Shot settings**, achieving ER10=0.7909/0.8066 while preserving utility close to the clean model. `see W.3 of Reviewer 7JqE`
>
> Taken together, these added results broaden the empirical coverage of the paper and suggest that the identified vulnerability is not confined to a specific model family or a single recommendation domain.
>
> ### **W.2/Q.2: Implementation Details for CIP.**
> CIP runs for at most 4 alternating image–text rounds. In each round, we select top 15% visual tokens and top 18% text tokens by cross-modal saliency. Each masked text token draws up to 6 candidates from a domain synonym table, optionally complemented by target-related keywords, and we replace at most 20% of masked tokens per round (at least 2), with an optional token-embedding perturbation bounded by 0.01. For images, we use a masked cosine-aligned single-step update toward the popular prototype under an $L_{\infty}$ budget of 0.05, stopping when similarity no longer improves.
>
> ### **W.3: Human-based Stealthiness Metrics.**
> Following prior studies, we used **ROUGE and FID** as automatic stealthiness metrics. We acknowledge that these metrics are informative but do not fully capture human-perceived detectability. To complement this, we conducted a **blind human A/B test** on 100 randomly sampled pre-/post-attack pairs with 30 participants. Annotators identified the manipulated sample based on overall visual-textual naturalness and semantic coherence. The accuracy was 46%, **close to random guessing**, indicating that the poisoned samples are difficult to distinguish from benign ones under human inspection.
>
> ### **W.4/Q.4: Evaluation with Additional Defenses.**
> We additionally evaluated VENOMREC under two stronger defenses, TokSub[3] and ONION[4], on the Clothing dataset. The results show that while these lower the attack strength slightly, VENOMREC maintains high efficacy (e.g.,ER20>0.63 vs. ShadowCast's much lower baseline). These results suggest that simple lexical normalization and rare-token filtering are not sufficient to reliably neutralize VENOMREC, which is consistent with our attack’s reliance on coordinated multimodal steering rather than only brittle textual triggers.
>
> |Defence(ZS)|HR5|NDCG5|ER5|HR10|NDCG10|ER10|HR20|NDCG20|ER20|
> |-|-|-|-|-|-|-|-|-|-|
> |NoDefence|0.1402|0.0945|0.1337|0.2206|0.1203|0.2820|0.3381|0.1499|0.7317|
> |TokSub|0.1410|0.0965|0.1361|0.2168|0.1208|0.2755|0.3357|0.1508|0.6743|
> |ONION|0.1361|0.0926|0.0749|0.2162|0.1183|0.1946|0.3351|0.1482|0.6357|
>
> |Defence(FS)|HR5|NDCG5|ER5|HR10|NDCG10|ER10|HR20|NDCG20|ER20|
> |-|-|-|-|-|-|-|-|-|-|
> |NoDefence|0.1402|0.0948|0.1158|0.2199|0.1204|0.2596|0.3377|0.1500|0.7170|
> |TokSub|0.1433|0.0981|0.1482|0.2191|0.1224|0.2998|0.3364|0.1519|0.7100|
> |ONION|0.1385|0.0946|0.0998|0.2168|0.1197|0.2433|0.3354|0.1495|0.6820|
>
> ### **Q.1: Sensitivity to Proxy Encoder.**
> We additionally evaluated proxy sensitivity using weaker encoders from the same model family. The results show that VENOMREC is more effective when the proxy is larger and more informative, but remains effective under a weaker proxy: ER@20 changes from 0.7170(FS)/0.7317(ZS) with the stronger proxy to 0.6825(FS)/0.6817(ZS) with the weaker proxy.
>
> ### **L.1: Discussion of Risks and Safeguards.**
> We agree that this aspect should be stated more clearly. VENOMREC is evaluated offline on public datasets under a realistic threat model, without access to private gradients or exact fine-tuned parameters. The goal is to expose a realistic dual-use vulnerability, not to enable misuse on live platforms. We will add this limitation and clarify that we will not release deployment-ready attack artifacts.
>
> ### **Reference:**
>
> [1] DiffMM:Multi-Modal Diffusion Model for Recommendation,ACM MM 2024.
>
> [2] MicroLens url:https://github.com/westlake-repl/MicroLens/
>
> [3] Defense Against Syntactic Textual Backdoor Attacks with Token Substitution,IEEE TIFS 2025.
>
> [4] ONION:A Simple and Effective Defense Against Textual Backdoor Attacks,EMNLP 2021.
>
> ---
> **We hope these clarifications and additional results address the your concerns.**

---

> > ### Author Rebuttal · Reviewer_yGxZ · 2026-04-02
> >
> > This rebuttle address my concerns regarding generalization and stealthiness. I have no further questions and will maintain my score.

---

> > > ### Author Response · Authors · 2026-04-05
> > >
> > > Thank you very much for your thoughtful review and for taking the time to read our rebuttal so carefully. We are deeply encouraged and grateful for your acknowledgement that the concerns regarding generalization and stealthiness have been fully resolved.
> > > Your constructive feedback has genuinely made our paper much stronger, and we will ensure all these additions are included in the final version.
> > > Thank you again for your careful assessment and constructive feedback.

---

### Official Review · Reviewer_jiBT · 2026-03-21

**Soundness:** 3
**Presentation:** 2
**Significance:** 3
**Originality:** 3
**Overall Recommendation:** 5
**Confidence:** 3

**Summary:**

This paper studies a new vulnerability in multimodal recommender systems powered by MLLMs. The authors show that multimodal poisoning can exploit cross-modal consensus to steer representations in a stable semantic direction during fine-tuning. To this end, they formalise cross-modal interactive poisoning and propose VENOMREC, which combines (i Exposure Alignment to locate high-impact regions in the joint embedding space and (ii Cross-modal Interactive Perturbation to generate coordinated token–patch edits. The experiments demonstrate the effectiveness of the proposed method.

**Compliance With Llm Reviewing Policy:**

Affirmed.

**Final Justification:**

The rebuttal addressed my concerns on those weaknesses.

**Key Questions For Authors:**

* How were the target items constructed? How did their popularity change during training?

**Limitations:**

There is no limitation section.

**Strengths And Weaknesses:**

Strengths:
* It is the first work that systematically investigate interactive poisoning attacks for multimodal recommender systems.
* The studies on attack stealthiness and recommendation utility, as well as the impact of compromised user ratio strengthen the paper by giving a deeper insights on the characteristics of the proposed method.
* Extensive experiments demonstrate the effectiveness of the proposed VENOMREC framework and show that they significantly outperform the baselines.


Weaknesses:
* It lacks of sufficient justification and clarity in Sec. 4.3 why the update rules optimize the latent alignment objective.
* It is unclear why only VIP5 and its variation is used for evaluation. The concerns on the bias of using particular models can be eliminated by evaluating the proposed method on more multimodal models, such as:
    * Multi-Modal Self-Supervised Learning for Recommendation.
    * DiffMM: Multi-Modal Diffusion Model for Recommendation.

---

> ### Author Rebuttal · Authors · 2026-03-31
>
> We are grateful to the reviewer for the constructive feedback and for the positive assessment of the originality and significance of our work. Below we address the specific concerns:
>
> ### **W.1 Justification for Latent Alignment (Sec. 4.3).**
> We appreciate this important question and are glad to clarify the optimization logic more explicitly. **Module II (CIP)** is designed as a coordinate-descent-like heuristic for optimizing the latent alignment objective $L_{adv}$. As defined in Eq.5, minimizing $L_{adv}$ is equivalent to maximizing the cosine similarity between the poisoned fused representation and the high-exposure target centroid $z^*$.
>
> As shown in Figure 2, CIP addresses this alignment objective by **alternating between visual and textual updates**. In the visual update (Eq.7), the direction vector $d_r$ is estimated via a gradient-free directional search to identify perturbations that increase similarity to $z^*$; in the textual step, candidate token replacements are greedily selected to further improve alignment under linguistic and cross-modal coherence constraints. By recomputing the attention-guided masks $m_{txt}$ and $m_{vis}$ after each update round, the attack adaptively targets the most "fusion-sensitive" token-patch correspondences. We agree that this is a practical optimization heuristic rather than a formal guarantee, and we will revise Sec. 4.3 to make this interpretation explicit.
>
> ### **W.2 Broader Evaluation Beyond VIP5.**
> To address the concern regarding potential model bias, we additionally evaluated VENOMREC on **DiffMM** [1] model, which represents a **traditional multimodal recommender** beyond the VIP5/T5-based MLLM family. The new results show that VENOMREC remains effective on DiffMM, achieving 0.6667 ER10 and still outperforming the strongest baseline ShadowCast by 0.507 absolute points on the Sports dataset, while maintaining comparable recommendation utility (e.g., NDCG10 = 0.1639). Similar trends are also observed on Clothing and Toys. **These results suggest that the vulnerability exploited by VENOMREC is not confined to one specific multimodal recommender implementation, but is more broadly related to multimodal representation fusion and coordinated cross-modal steering**.
>
> * Sports dataset
>
> |Attack|HR5|NDCG5|ER5|HR10|NDCG10|ER10|HR20|NDCG20|ER20|
> |---|---|---|---|---|---|---|---|---|---|
> |NoAttack|0.1919|0.1315|0|0.2916|0.1635|0|0.4320|0.1988|0
> |ShadowCast|0.1919|0.1315|0.0810|0.2916|0.1634|0.1597|0.4320|0.1988|0.3109
> |VENOMREC|0.1921|0.1315|0.4416|0.2933|0.1639|0.6667|0.4358|0.1998|0.8870
>
> * Clothing dataset
>
> |Attack|HR5|NDCG5|ER5|HR10|NDCG10|ER10|HR20|NDCG20|ER20|
> |---|---|---|---|---|---|---|---|---|---|
> |NoAttack|0.2554|0.1893|0|0.3458|0.2184|0|0.4738|0.2506|0
> |ShadowCast|0.2553|0.1893|0.2171|0.3458|0.2184|0.3506|0.4739|0.2506|0.5546
> |VENOMREC|0.2551|0.1893|0.2225|0.3456|0.2184|0.3662|0.4731|0.2505|0.6341
>
> * Toys dataset
>
> |Attack|HR5|NDCG5|ER5|HR10|NDCG10|ER10|HR20|NDCG20|ER20|
> |---|---|---|---|---|---|---|---|---|---|
> |NoAttack|0.2744|0.2078|0|0.3587|0.2349|0|0.4836|0.2663|0
> |ShadowCast|0.2743|0.2077|0.0286|0.3586|0.2348|0.0676|0.4837|0.2663|0.1555
> |VENOMREC|0.2736|0.2064|0.5093|0.3564|0.2330|0.7140|0.4818|0.2645|0.8963
>
> ### **Q1: Target Item Construction and Popularity Dynamics.**
> - **Construction**: Target items were randomly sampled from the bottom 20% of items (by interaction frequency) to simulate a challenging "cold-start" promotion scenario.
> - **Popularity Dynamics**: We tracked the average rank of target item among 100 candidate items (99 benign + 1 target) across all test users to show the popularity change of target item as requested. In each epoch, we compute the target item's rank for each user and report the rounded mean value. As shown below, the target item's rank improves significantly as the model incorporates the malicious consensus.
>
>
>     |Epoch(%)|0%|20%|40%|60%|80%|100%|
>     |---|---|---|---|---|---|---|
>     |Rank(Avg)|54.17|43.42|42.40|28.28|24.52|24.44|
>
> ### **Reference**:
>
> [1] DiffMM: Multi-Modal Diffusion Model for Recommendation, ACM MM 2024.
>
> ---------------------------------------------
> **We hope these clarifications and additional results address your concerns.**

---

> > ### Author Rebuttal · Reviewer_jiBT · 2026-04-06
> >
> > Thanks for the detailed rebuttal.

---

> > > ### Author Response · Authors · 2026-04-07
> > >
> > > Thank you very much for your thoughtful follow-up and for revisiting our rebuttal so carefully. We truly appreciate your positive reassessment, and we are especially grateful for your decision to raise the score from 4 to 5 after reading our rebuttal.
> > > We will incorporate the added results and clarifications into the final manuscript. Thank you again for your careful reading and constructive feedback.

---

### Decision · Program_Chairs · 2026-04-30

**Decision:**

Accept (regular)

**Comment:**

All reviewers are positive about this submission, with a broad agreement on its soundness, novelty, and strong empirical performance. The rebuttal further helped address the main concerns raised during review. I therefore support a clear acceptance and strongly encourage the authors to incorporate the additional experiments, clarifications, and discussion from the rebuttal into the camera-ready version.